

## 2 Designing global climate and atmospheric chemistry simulations for 1 km and 10 km diameter asteroid impacts using the properties of ejecta from the K-Pg impact

Owen B. Toon[1], Charles Bardeen[2], Rolando Garcia[2]
[1] Department of Atmospheric and Oceanic Science, Laboratory for Atmospheric and
Space Physics, University of Colorado, Boulder
[2] National Center for Atmospheric Research, Boulder, Colorado
*Correspondence to:* O.B. Toon (toon@lasp.colorado.edu)
**Abstract.** About 66 million years ago an asteroid about 10 km in diameter struck the
Yucatan Peninsula creating the Chicxulub crater. The crater has been dated and found to
be coincident with the Cretaceous-Paleogene (K-Pg) mass extinction event, one of 6 great
mass extinctions in the last 600 million years. This event precipitated one of the largest
episodes of rapid climate change in Earth history, yet no modern three-dimensional
climate calculations have simulated the event. Similarly, while there is an on-going effort
to detect asteroids that might hit Earth and to develop methods to stop them, there have
been no modern calculations of the sizes of asteroids whose impacts on land would cause
devastating effects on Earth. Here we provide the information needed to initialize such
calculations for the K-Pg impactor and for a 1 km diameter impactor.
There is considerable controversy about the details of the events that followed the
Chicxulub impact. We proceed through the data record in the order of confidence that a
climatically important material was present in the atmosphere. The climatic importance
is roughly proportional to the optical depth of the material. Several hundred-micron
diameter spherules are found globally in an abundance that would have produced an
atmospheric layer with an optical depth around 20, yet their large sizes would only allow
them to stay airborne for a few days. They were likely important for triggering global
wildfires. Soot, probably from global or near-global wildfires, is found globally in an
abundance that would have produced an optical depth near 100, which would effectively
prevent sunlight from reaching the surface. Nanometer sized iron particles are also
present globally. Theory suggests these particles might be remnants of the vaporized
asteroid and target that initially remained as vapor rather than condensing on the
hundred-micron spherules when they entered the atmosphere. If present in the abundance
suggested by theory, their optical depth would have exceeded 1000. Clastics may be
present globally, but only the quartz fraction can be quantified since shock features can
identify it. However, it is very difficult to determine the total abundance of clastics. We
reconcile previous widely disparate estimates and suggest the clastics may have had an
optical depth near 100. Sulfur is predicted to originate about equally from the impactor
and from the Yucatan surface materials. By mass, sulfur is less than 10 percent of the
mass of the spheres and nano-particles. Since the sulfur probably reacted on the surfaces
of the soot, nano-particles, clastics and spheres, it is likely a minor component of the
climate forcing; however, detailed studies of the conversion of sulfur gases to particles
are needed to determine if sulfuric acid aerosols dominated in late stages of the evolution





of the atmospheric debris. Numerous gases, including $CO_2$, $SO_2$ (or $SO_3$), $H_2O$, $CO_2$, Cl,
Br, and I, were likely injected into the upper atmosphere by the impact or the immediate
effects of the impact such as fires across the planet. Their abundance might have
increased relative to current ambient values by a significant fraction of current values for
$CO_2$, and by factors of 100 to 1000 for the other gases.
For the 1 km impactor, nano-particles might have had an optical depth of 1.5 if the
impact occurred on land. If the impactor struck a densely forested region, soot from the
forest fires might have had an optical depth of 0.1. Only S and I would be expected to be
perturbed significantly relative to ambient gas phase values. 1 km asteroids impacting
the ocean may inject seawater into the stratosphere as well as halogens that are dissolved
in the seawater.
For each of the materials mentioned we provide initial abundances and injection altitudes.
For particles we suggest initial size distributions and optical constants. We also suggest
new observations that could be made to narrow the uncertainties about the particles and
gases generated by large impacts.

**Keywords** Climate modeling; Initial conditions; Asteroid impacts; K-Pg extinction

**1. Introduction and definitions**
About 66 million years ago an asteroid around 10 km in diameter hit the Earth near the
present day Yucatan village of Chicxulub and created an immense crater whose age
coincides with the Cretaceous-Paleogene (K-Pg) global mass extinction (Alvarez et al.,
1980; Schulte et al., 2010; Renne et al., 2013). There is an enormous literature
concerning this event and its aftermath. Surprisingly, however, there are very few papers
about the changes in climate and atmospheric chemistry caused by the debris from the
impact while it was in the atmosphere, and no studies based on modern three-dimensional
climate models. Nevertheless, this event was almost certainly one of the largest and most
dramatic short-term perturbations to climate and atmospheric chemistry in Earth history.
There is substantial evidence for many other impacts in Earth history as large or larger
than that at Chicxulub, mostly in the Pre-Cambrian (e.g. Johnson and Melosh, 2012a;
Glass and Simonson, 2012). There is also a growing effort to find asteroids smaller than
the one that hit Chicxulub, but whose impact might have significant global effects, and to
develop techniques to stop any that could hit the Earth. For example, as of November 17,
2015 NASA's Near Earth Object Program identifies 13,392 objects whose orbits pass
near Earth. Among these objects, 878 have a diameter of about 1 km or larger, and 1640
have been identified as Potentially Hazardous Asteroids, which are asteroids that pass the
Earth within about 5% of Earth's distance from the sun, and are larger than about 150 m
diameter.
There is evidence for such smaller impacts in recent geologic history from craters,
osmium variations in sea cores (Paquay et al., 2008), and spherule layers (Johnson and
Melosh, 2012a; Glass and Simonson, 2012). For instance, a multi-kilometer object
formed the Siberian Popigai crater in the Late Eocene and another multi-kilometer object



formed the Late Eocene Chesapeake Bay crater in the United States. Size estimates vary
between techniques, but within a given technique the Popigai object is generally given a
diameter half that of the Chicxulub object. Toon et al. (1997) point out that the effects of
impacts scale with the impactor energy, or cube of the diameter, not diameter (or crater
size). The Popigai object likely had about 12% of the energy of the Chicxulub object.
Surprisingly, except for collisions in the ocean (Pierazzo et al., 2012), climate models
have not been used to determine the destruction that might be caused by objects near 1
km in diameter, a suggested lower limit to the size of an impactor that might do
significant worldwide damage (e.g. Toon et al., 1997).
Here we describe the parameters that are needed to initialize three-dimensional climate
and atmospheric chemistry models for the Chicxulub impact and for a 1-km diameter
asteroid impact. Nearly every aspect of the K-Pg impact event is uncertain, and
controversial. We will address some of these uncertainties and controversies and make
recommendations for the initial conditions that seem most appropriate for a climate
model, based upon the geological evidence. We will also suggest the properties of the
initial impact debris from a 1 km diameter asteroid.
There are numerous observed and predicted components of the Chicxulub impact debris.
The distal debris layer, defined to be the debris that is more than 4000 km removed from
the impact site, is thought to contain material that remained in the atmosphere long
enough to be globally distributed. This distal layer, sometimes called the fireball layer or
the magic layer, is typically only a few mm thick (Smit, 1999). As discussed below, the
layer includes 200 $\mu$m-sized spherules, 50 $\mu$m-sized shocked quartz grains, 0.1-$\mu$m-sized
soot and a 20 nm-sized iron-rich material.
We discuss each of the components of the distal layer in detail below. As an outline of
this discussion we find the following: The large spherules are not likely to be of
importance to the climate because they would have been removed from the atmosphere in
only a few days. However, they may have initiated global wildfires. The shocked quartz
grains, one of the definitive pieces of evidence for an impact origin as opposed to
volcanic origin of the debris layer, is likely only a small fraction of the clastic debris. It
is difficult to identify the rest of the minerals produced by crushing because there is
material in the layer that might have been produced long after the impact by erosion and
chemical alteration of the large spheres or from the ambient environment. One major
controversy surrounding the clastic material is the fraction that is submicron-sized.
Particles larger than a micron will not remain in the atmosphere very long and, therefore,
are less likely to affect climate. Unfortunately, the sub-$\mu$m micron portion of the clastics
in the distal layer, which might linger in the atmosphere for a year or more, has not been
directly measured. Our estimate of the mass of submicron-sized clastics suggest that it
could have had a very high large optical depth that would be capable of modifying the
climate significantly. Nevertheless, submicron clastics are only of modest climatic
importance relative to the light absorbing soot and possibly the iron rich nm-scale debris.
Submicron soot is observed in the global distal layer in such quantity that it would have
had a very great impact on the climate when it was suspended in the atmosphere. The
major controversy surrounding the soot is whether it originated from forest fires, or from
hydrocarbons at the impact site. The origin of the soot, however, is of secondary
importance with regard to its effect on climate. Since the soot layer overlaps the iridium



layer in the distal debris it had to have been created within a year or two of the impact,
based on the removal time of small particles from the atmosphere (and ocean), and could
not have been the result of fires long after the impact. The fireball layer is often colored
red and contains abundant iron. Some of the iron has been identified as part of a 20 nm-
sized particle phase, possibly representing a portion of the recondensed vaporized
impactor and target. However, relatively little work has been done on this material. Its
abundance has not been measured, but theoretical work suggests its mass could have been
comparable to that of the impactor. Therefore, the nm-sized particles could have been of
great importance to the climate. Each of the materials just described is present in the
distal layer, and their impacts on the atmosphere were likely additive.
There are several other possible components of the distal layer that have not been clearly
identified and studied as part of the impact debris, which we discuss below. Water,
carbon, sulfur, chlorine, bromine, and iodine were likely present in significant quantities
in the atmosphere after the impact. The Chicxulub impact occurred in the sea with depths
possibly ranging up to 1 km. The target sediments and the asteroid probably also
contained significant amounts of water. Water is an important greenhouse gas, and could
condense to form rain, which might have removed materials from the stratosphere.
Carbon is present in seawater, in many asteroids and in sediments. Injections as carbon
dioxide or methane might have led to an increased greenhouse effect. Sulfur is widely
distributed in the ambient environment, and is water-soluble. Therefore, it is difficult to
identify extraterrestrial sulfur in the debris layer. However, the impact site contains a lot
of sulfur, and asteroids also contain significant amounts of sulfur. Sulfur is noteworthy
because it is known to produce atmospheric particulates in today's atmosphere that alter
the climate. Chlorine, bromine and iodine can destroy ozone, and their effectiveness as
catalysts is enhanced by heterogeneous reactions on sulfuric acid aerosols.
In addition to the mm-thick distal layer, there is an intermediate region ranging from
2,500-4,000 km from the impact site with a debris layer that is several cm thick (Smit,
1999). This layer contains microtektites (molten rock deformed by passage through the
air), shocked quartz, as well as clastics such as pulverized and shocked carbonates. Most
of this layer originated from the target material in the Yucatan. It is of interest because,
like the debris clouds from explosive volcanic eruptions, components of this material
may have escaped from the region near the impact site to become part of the global debris
layer.
Properties of each of these materials need to be known in order to model their effects on
the climate and atmospheric chemistry realistically. These properties include the altitude
of injection, the size of the injected particles, the mass of injected particles or gases, the
density of the particles, and the optical properties of the injected particles and gases. Our
best estimates for these properties for the K-Pg impact are summarized in Table 1 for
particles and Table 2 for gases, and discussed for each material in Section 2. Tables 3
and 4 provide an extrapolation of these properties for an impact of by a 1 km sized object.
While the mass of the injected material is useful as an input parameter to a model, the
optical depth of the particles is needed to quantify their impact on the atmospheric
radiation field and, therefore, on the climate. Hence, optical depth is a useful quantity to
compare the relative importance of the various materials to the climate. For a



monodisperse particle size distribution, the optical depth is given by $\tau = \dfrac{3Mq_{ext}}{4\rho r}$. Here $M$
is the mass of particles in a column of air (for example, g cm$^{-2}$), $r$ is the radius of the
particles, $\rho$ is the density of the material composing the particles, and $q_{ext}$ is the optical
extinction efficiency at the wavelength of interest. The optical extinction efficiency is a
function of the size of the particles relative to the wavelength of light of interest, and of
the optical constants of the material. The optical extinction efficiency is computed
accurately in climate models. However, a rough value of $q_{ext}$ for particles larger than 1
$\mu$m, is about 2 for visible wavelength light. We use this rough estimate for $q_{ext}$ in Table
1 and Table 3 to calculate an optical depth for purposes of qualitatively comparing the
importance of the various types of injected particles. We assume in the heuristic
calculations of optical depth in Tables 1 and 3 that the particles have a radius of 1$\mu$m
because smaller particles will quickly coagulate to a radius near 1 $\mu$m given the large
masses of injected material.
Below we define the properties that are needed to perform climate or atmospheric
chemistry simulations for each material that might be important.

**2. Particulate Injections**
**2.1 Large spherules**
**2.1.1 Large spherules from the Chicxulub impact**
The most evident component of the distal and regional debris layers is spherical particles,
some of which are large enough to be seen with the naked eye. Due to their spherical
shape it is assumed that they are part of the melt debris from the impact or the condensed
vapor from the impact (Johnson and Melosh, 2012b; 2014). The particles are not thought
to have melted on reentry into the atmosphere since debris launched above the
atmosphere by the impact should not reach high enough velocities to melt when it
reenters the atmosphere. According to Bohor and Glass (1995) there are two types of
spherules, with differing composition and distribution. They identify Type 1 splash-form
spherules (tektites or microtektites) that occur in the melt-ejecta (basal or lower) layer of
the regional debris layer where it has a two-layered structure. These spherules are found
as far from the Chicxulub site as Wyoming, but generally do not extend beyond about
4000 km away from Chicxulub. While the type 1 particles are derived from silicic rocks,
they are also mixed with sulfur rich carbonates from the upper sediments in the Yucatan.
The Type 1 spherules are poor in Ni and Ir, and the lower layer is poor in shocked quartz,
consistent with their origin from the low energy impact ejecta from the crater. Generally
the debris layer within about 4000 km of the crater is almost entirely composed of target
material, rather than material from the impactor itself. Type 2 spherules, on the other
hand, are found in the distal debris layer, and presumably formed primarily from the
condensation of rock vapor from the impactor and target (O'Keefe and Ahrens, 1982;
Johnson and Melosh, 2012b). There are sub-types of Type 2 spherules that correspond to
varying composition of the original source material. Type 2 spherules occur in the upper
layer in impact sites near Chicxulub, which merges into the fireball layer at distal sites.



The Type 2 spherules are rich in Ni and Ir, while the fireball layer is rich in shocked
quartz.
The formation of the spherical particles may depend on two different processes. Melosh
and Vickery (1991) describe one formation mechanism, probably occurring in less
heavily shocked portions of the target, when molten material decompresses until it
reaches a critical line at which it starts to boil. The gas drag from the rock vapor on the
molten rock spheres then tears apart the molten material, just as water droplets break
apart when they fall through air. The relative velocities of water drops in air and the melt
in vapor are similar, as are the surface tensions. As a result melt droplets are similar in
size to drizzle drops in light rain, near 250 $\mu$m. According to Johnson and Melosh
(2012b) these spherical particles are most likely to be found within 4000 km of the
impact site, and to be chemically related to the target material, and not to the impactor.
Such materials are reported across North America as Type 1 spherules (Bohor et al.,
1987), and are sometimes referred to as microtektites. Since these spherules are not
global, they likely were not as relevant to climate as the Type 2 spherules.
Melt droplets can also form in heavily shocked parts of the impact debris as rock vapor
condenses to form melt in the fireball, which rises thousands of km above the Earth's
surface. These melt droplets form the Type 2 spherules. O'Keefe and Ahrens (1982) first
modeled this process, and deduced that particles near a few hundred microns in size
would form, as is observed. They also pointed out that the size of the spheres would be
proportional to the size of the impactor. Johnson and Melosh (2012b) recently
reconsidered this process for forming melt particles. They point out that the large
spherules contain iridium (e.g., Smit, 1999), which is consistent with them being
composed partially of the vaporized impactor. Their model of the formation and
distribution of these particles suggests the particles have a size that varies spatially over
the plume. Averaging over the plume yields a mean size of 217 $\mu$m with a standard
deviation of about 47 $\mu$m for a 10 km diameter impactor hitting at 21 km s$^{-1}$. From the
two examples given by Johnson and Melosh (2012b) it appears that the standard
deviation is consistently 22% of the mean radius for asteroids of different sizes. The
initial values for the various properties of Type 2 spherules described above are
summarized in Table 1 for the K-Pg impactor.
Smit (1999), who refers to the Type 2 spherules in the distal layer as microkrystites,
estimated that these particles typically have a diameter near 250 $\mu$m, and a surface
concentration of about 20,000 particles cm$^{-2}$ over the Earth. Unfortunately, we are not
aware of studies that measure the dispersion of the size distribution, or the spatial
variation of the abundance of these particles. We assume that the particles have the
density of CM2 asteroids, since Cr isotope ratios suggest that is the composition of the K-
Pg impactor (Trinquier et al., 2006). Assuming this density, ~2.7 g cm$^{-3}$, the mass of
spherules per unit area of the Earth is about 0.4 g cm$^{-2}$, and the initial optical depth is
about 20, as noted in Table 1. These spherules compose about half of the mass of the
distal layer. We assume the particles were initially distributed uniformly around the
globe, with the initial mixing ratio in the atmosphere varying only in altitude. Some
theoretical studies, such as Kring and Durda (2002) and Morgan et al. (2013), suggest





that these particles were not uniformly deposited in latitude and longitude, but had
focusing points such as the antipodes of the impact site. Unfortunately, we are not aware
of quantitative data on the global distribution of the spherules. The study by Morgan et
al. (2013) may also be more applicable to the Type 1 spherules since their numerical
model does not produce vaporized impactor.
According to the simulations of Goldin and Melosh (2009), the in-falling spherical
particles reached terminal fall velocity at ~70km altitude, at which point they begin to
behave like individual airborne particles. Kalasnikova et al. (2000) investigated
incoming micrometeorites in the present atmosphere, which generally ablate near 85 km.
Kalasnikova et al. (2000) find material entering from space stops in the atmosphere after
it encounters a mass of air approximately equal to its own mass. Therefore, the altitude
distribution is taken to be Gaussian, centered at 70 km and with a half-width of one
atmospheric scale height (about 6.6 km based on the U.S. Standard Atmosphere). A scale
height is chosen as the half width of the injection profile since it is a natural measure of
the density of the atmosphere. Figure 1 illustrates the vertical injection profile of the
spherules (green curve). As discussed below we expect several materials with origins
similar to those of the spherules to be injected in this same altitude range, but others with
origins unrelated to the impact generated plume, such as soot from fires, to be injected at
lower altitudes.
The energy release from the reentry of these large spherical particles into the atmosphere
was likely responsible for setting most of the above ground terrestrial biosphere on fire.
However, due to their size, the spherules could not have remained in the atmosphere for
more than a few days. Hence they likely did not have a significant direct impact on the
climate, but fell to Earth like a gentle rain.

### 287 2.1.2 Large spherules from a 1 km diameter asteroid impact

Like O'Keefe and Ahrens (1982), Johnson and Melosh (2012b) conclude that the particle
size will vary in proportion to the impactor diameter. For a 1 km diameter impactor
hitting the land they suggest that the mean diameter of the spherical particles will be
about 15 $\mu$m. Table 3 provides our assumed properties of the spherules from a
hypothetical 1 km diameter impactor hitting the land. It is likely that spherules would be
distributed over much of the globe even for the 1 km diameter impact. Johnson and
Melosh (2012a) as well as Glass and Simonson (2012) report a spherule layer associated
with the Popagai impact in the late Eocene. This layer contains spherules similar in size
or even larger than those associated with the Chicxulub impact. However, this layer is
only about 10% as thick as the distal layer from the Chicxulub impact. A 1 km impactor
hitting the deep oceans may not produce a layer of spherules.

### 300 2.2 Soot

### 301 2.2.1 Soot from the Chicxulub impact

Spherical soot (also referred to as black carbon, or elemental carbon) particles were
discovered in the boundary layer debris at sites including Denmark, Italy, Spain, Austria,
Tunisia, Turkmenistan, the United States and New Zealand by Wolbach et al. (1985;




1988; 1990). Soot was also found in anaerobic deep-sea cores from the mid-Pacific
(Wolbach et al., 2003). Soot was apparently lost by oxidation in aerobic deep-water sites
in the 66 million years since emplacement. There is debate about whether these particles
originated from global wildfires, or from the impact itself (Belcher et al., 2003, 2004,
2005, 2009; Belcher, 2009; Harvey et al., 2008; Robertson et al., 2013a, Pierazzo and
Artemieva (2012), Premovic (2012), Morgan et al. (2013)). Robertson et al. (2013) and
the other more recent papers, argue that it is implausible that there was enough carbon at
the impact site to produce the amount of soot observed by Wolbach et al. (1988). This
debate about the origin of the particles does not greatly affect the effect these particles
would have had on the climate when they were suspended in the atmosphere. The
particles are small and widely distributed, and so must have remained in the atmosphere
for a few years. They are numerous and so must have produced a very large optical depth
and, being composed of carbon, they would have been excellent absorbers of sunlight.
Whether the soot particles originated from global fires and were deposited in the upper
troposphere, or they originated at the impact site and were deposited in the mesosphere,
the climate effect of the observed soot would have been very great. Some have suggested
that the soot resulted from wildfires in dead and dying trees that occurred well after the
impact. However, Wolbach et al. (1988) show that soot and iridium are tightly correlated
and collocated. The soot and iridium in the distal layer must have been deposited within
a few years of the impact, since small particles will not stay in the air much longer.
Therefore, any fires must have been very close in time to the impact.
Wolbach et al. (1988) estimated the global mass of soot in the debris layer as $7\pm4$ x $10^{16}$ g
of C or equivalently $1.3 \times 10^{-2}$ g C cm$^{-2}$. This mass of soot would require that the bulk of the
above ground biomass burned and was partially converted to soot with an efficiency of 3%,
assuming the biomass is 1.5 g C cm$^{-2}$ of above ground, dry organic mass per cm$^2$ over the
land area of Earth, which is typical of current tropical forests. This inferred 3% emission
factor is about 60 times greater than that suggested by Andreae and Merlet (2001) for
current wildfires, but agrees with laboratory and other observations from burning wood
under conditions consistent with mass fires (Crutzen et al., 1984; Turco et al., 1990). The
high soot emission efficiency inferred for the K-Pg impact likely represents the processes
occurring in firestorms, also called mass fires, set globally after the impact as opposed to
the processes observed in typical forest fires and discussed by Andreae and Merlet (2001).
Mass fires are more intense than forest fires, and consume all the fuel available, possibly
including that in the near surface soil. Ivany and Salawitch (1993) argued independently
from oceanic carbon isotope ratios that at least 25% of the above ground biomass must
have burned at the K-Pg boundary.
As noted in Table 1, the mass of soot found by Wolbach et al. (1988) would produce an
optical depth near 100 if the particles coagulated to spheres with a radius of 1 $\mu$m while
they were in the atmosphere. Toon et al. (1997) pointed out that soot clouds with such a
large optical depth would reduce light levels at the Earth's surface effectively to zero. The
optical and chemical evolution of the particles once in the atmosphere may be influenced
by the presence of liquid organics on the soot particles. Bare soot particles coagulate into
chains and sheets, while particles that are coated by liquids may form balls. Chains,
sheets, and coated balls have very different optical properties than do spheres (Wolf and



Toon, 2010; Ackerman and Toon, 1981; Bond and Bergstrom, 2006; Mikhailov et al.,
2006). Particulate organic matter can be absorbing, and soot coated with organics can
have enhanced absorption relative to soot that is uncoated (Lack et al., 2012; Mikhailov
et al., 2006). These fractal shapes, and organic coatings might not be preserved in samples
in the distal layer since all the particles have been consolidated in a layer, and even in the
current atmosphere the organics have short lifetimes due to rapid oxidation.

Wolbach et al. (1985) fit the size of the particles they observed, after exposing them to
ultrasound to break up agglomerates, to a lognormal size distribution, described by

$$\frac{dN}{d\ln r} = \frac{N_t}{\ln\sigma\sqrt{2\pi}}\exp[-(\ln^2(\frac{r}{r_m})/2\ln^2\sigma)] \ . \tag{1}$$


Here $r$ is the particle radius, $N_t$ is the total number of particles per unit volume of air, $r_m$ is
the mode radius, and $\sigma$ is the width of the distribution. Wolbach et al. (1985) find $r_m =$
0.11 $\mu$m, and $\sigma = 1.6$ for the soot in the K-Pg boundary layer. We assume this
distribution represents the initial sizes of the soot particles. The final size, which would
be determined by coagulation while in the atmosphere, might not preserved in the
sediments, and loosely bound clumps of particles would have been destroyed by the
ultrasound treatment of the samples.

The size distribution of soot from the K-Pg boundary is similar to that of smoke nearby
present day biomass fires as indicated in Fig. 2 (e.g., Matichuk et al., 2008). This
similarity in sizes is somewhat surprising because the present day smoke size distribution
includes organic carbon, which is present in addition to the elemental carbon (soot).
Generally, in wildfire smoke organic carbon has 5-10 times the mass of soot, so one
might anticipate that the K-Pg soot would be about half the size of the present day smoke
rather than of similar size since the organic coatings are no longer present, or were never
present, on the K-Pg soot. The organics might never have been present, because mass
fires are very intense and tend to consume all the available fuel, which might include the
organic coatings. Aggregation in the hot fires may have caused this slightly larger than
expected size in the K-Pg sediments. Wolbach et al. (1985) suspended their samples in
water and subjected them to ultrasound for 15 minutes in a failed attempt to completely
break up agglomerates. This failure indicated that the remaining agglomerates might
have been flame-welded. Therefore, the K-Pg size distribution from Wolbach et al.
(1985) does not represent the monomers in the aggregate soot fractal structures. Rather
the K-Pg size distributions represent a combination of monomers and aggregates that may
have formed at high temperatures. Possibly the smallest sized particles measured by
Wolbach et al. (1985), which have radii of 30-60 nm, represent the soot monomers.
These are in the same general range as monomer sizes observed in soot from
conventional fires (Bond and Bergstrom, 2006).

The injection altitude of the soot depends on its source. In a series of papers Belcher et
al. (2003; 2004; 2005; 2009) and Belcher (2009) argued from multiple points of view that
there were no global forest fires, and Harvey et al. (2008) argued that the soot originated
from oil, coal and other organic deposits at the location of the impact. If correct, the soot



might have been injected at high altitude along with the large spherules. Recently,
Robertson et al. (2013a) reconsidered each of the arguments presented by Belcher et al.
and came to the conclusion that global wildfires did indeed occur.  Pierazzo and
Artemieva  (2012), Premovic (2012), Morgan et al. (2013), as well as Robertson et al.
(2013a) have independently argued that oil and other biomass in the crater is
quantitatively insufficient to be the source of the soot.  Therefore, we assume that the
soot indeed originated from burning biomass distributed over the globe.  The soot is
clearly present in the distal layer material, and therefore was once in the atmosphere
where it could cause significant changes to the climate.

Toon et al. (2007) have outlined the altitudes where one expects large mass fires to inject
their smoke.  Numerical simulations have shown that mass fires larger than about 5 km in
diameter have smoke cloud tops well into the stratosphere.  The smoke itself is
distributed over a range of heights, however. The details of the injection profiles depend
on the rate of fuel burning, the size of the fires, and the meteorological conditions among
other factors.  In addition, some smoke is quickly removed from the atmosphere by
precipitation in pyro-cumulus. However, it is thought that over-seeding of the clouds by
smoke prevents precipitation, and that only 20% or so of the smoke injected into the
upper atmosphere is promptly rained out (Toon et al., 2007).  Smoke that is injected near
the ground, on the other hand, will be removed by rainfall within days of weeks.

The K-Pg impact occurred at a time when average biomass density likely was higher than
now.  Following Small and Heikes (1988; Figure 3f) and Pittock et al. (1989) one would
expect smoke from large area fires burning in high biomass density areas to show a bi-
modal smoke injection profile.  The smoke at higher levels is injected in the pyro-
cumulus and other regions with strong vertical motions.  However, once the fires die-
down smoke will be emitted in the boundary layer.  There are also downdrafts, as well as
entrainment and mixing with the environment, that occur in all cumulus and these will
carry some smoke into the boundary layer. We simulate this with injections whose
vertical distributions are Gaussian functions centered at the tropopause and at the surface,
as illustrated in Fig. 1. The injection at the tropopause has a half width of 3 km, but
nothing is injected above about 25 km. We set this upper altitude limit based on the
heights of the stratospheric sulfate clouds from explosive volcanic eruptions, which rise
buoyantly as do smoke plumes. The Gaussian distribution at the ground has a half width
of 1 km, assuming that the local boundary layer is relatively shallow. We assume 50% of
the soot is contained in each of these distributions for the general case.  For the K-Pg, we
assume the soot observed in the distal layer was all in the portion of the Gaussian
distribution at the tropopause.
Therefore, the injection profile is given by:

$$I(g\ s^{-1}km^{-1}) = \frac{I_T}{2\sqrt{2\pi}}\left[\frac{1}{\mu}e^{\left(-0.5\left(\frac{z}{\mu}\right)^2\right)} + \frac{1}{\eta}e^{\left(-0.5\left(\frac{z-z_{trop}}{\eta}\right)^2\right)}\right] \quad (2)$$



Here $I$ is the mass emission rate per km of altitude, $I_T$ is the total mass emitted per second
after correcting for the emission altitude range (0-25 km) and grid spacing, $\mu$ is 1 km, $\eta$ is
3 km, and $z_{trop}$ is the altitude of the tropopause.
Geographically, we assume for the K-Pg event that all the surface biomass is set on fire.
For the 1 km diameter impact, however, only the region near the impact site would burn
as discussed further below.
There is also an issue of how long it takes to inject the smoke. Forest fires often burn for
days, advancing along a fire front as winds blow embers far beyond the flames and onto
unburned terrain. Mass fires may not spread because powerful converging winds restrict
the spread. However, little is known observationally about mass fires, and fires can
spread by intense infrared radiation lighting adjacent material. If mass fires are restricted
then they will burn only as long as they have fuel. The present above ground global
biomass in tropical forests is in the range of 0.6-1.2 g C cm$^{-2}$ (Houghton, 2005). The
energy content of biomass is on the order of $3 \times 10^4$ J/g C or, given the biomass
concentration just mentioned, about $3 \times 10^8$ J m$^{-2}$. Penner et al. (1986) and Small and
Heikes (1988) found that large area mass fires with energy release rates of 0.1 MW m$^{-2}$
would have plumes reaching the lower stratosphere. Hence, it would be necessary to
assume that the fuel burned in an hour or so to achieve these energy releases. Of course,
it might take some time for fires in different places to start fully burning, so considering
the entire region of the mass fire, as opposed to a small individual part of the fires, might
prolong the energy release considerably. For example, it took several hours for the mass
fire in Hiroshima to develop after the explosion of the atom bomb (Toon et al., 2007)
It should be noted that in simulations of stratospheric injections of soot from nuclear
conflicts, soot is self-lofted by sunlight heating the smoke (Robock et al., 2007b).
However, in the case of the K-Pg impact, if there are other types of particles injected
above the soot, which then block sunlight, the soot may not be self-lofted, which will
limit its lifetime. The initial soot distribution that is estimated here does not include the
effects of self-lofting, which would continue after the initial injection and should be part
of the climate simulation.
The final property to specify for soot is the optical constants. This issue is complicated
by the possible presence of organic material on the soot (Lack et al., 2012). However, it
is known that many of these organics are quickly oxidized by ozone, which is plentiful in
the ambient stratosphere. The stratosphere after the impact however, may have become
depleted in ozone very quickly, so that the organic coatings might have survived. It is
also possible that intense fires, such as mass fires, will consume the organic coatings,
which may explain why the production of soot in the fires seems to have been so much
more efficient than for normal fires. It may therefore be sufficient to treat the soot as
fractal agglomerates of elemental carbon (Bond and Bergstrom, 2006). It is known that
the optical properties of the agglomerates will not obey Mie theory. However, one may
treat their optical properties as well as their microphysical properties using the fractal
optics approach described by Wolf and Toon (2010). The optical constants for elemental
carbon may then be used for the monomers. Alternatively, one may add the organic mass



to the particles, and treat them using core-shell theory (Toon and Ackerman, 1981;
Mikhailov et al., 2006).
Bond and Bergstrom (2006) have thoroughly reviewed the literature on the optical
properties of elemental carbon. They conclude that the optical constants are most likely
independent of wavelength across the visible, with a value that depends on the bulk
density of the particles. Following their range of values for refractive index versus
particle density we suggest using a wavelength independent real index of refraction
$n$=1.80 and an imaginary index $k$=0.67. We also use these values in the infrared as
shown in Figure 3. For the monomers in Tables 1 and 3, we adopt the density suggested
by Bond and Bergstrom (2006) for light absorbing material, 1.8 g cm$^{-3}$.

### 496 2.2.2 Soot from a 1 km impact

Extrapolations of the soot injection parameters to smaller impactors than the one defining
the K-Pg boundary should only involve changes to the mass of soot injected, since the
basic properties of the soot at the K-Pg boundary are similar to those of forest fire soot.
Therefore, the particle sizes, injection heights, and optical constants recommended in
Table 3 for the smaller impact are the same as listed in Table 1 for the Chicxulub impact.
The mass of soot injected is estimated from the extrapolations in Toon et al. (1997). For
an impactor as small as 1 km diameter, debris from the impact site would not provide
sufficient energy to ignite the global biota since the energy of the 1 km impactor is about
1000 times less than that of the Chicxulub impactor. Instead, radiation from the ablation
of the incoming object and from the rising fireball at the impact site would ignite material
that is within visible range of the entering object and the fireball. This ignition
mechanism is well understood from nuclear weapons tests (Turco et al., 1990). Hence,
for a 1 km diameter impactor the fuel load at the site of the impact becomes critical to
evaluate the soot release. No soot would be produced from an impact in the ocean, an ice
sheet, or a desert. In Table 3 to compute the smoke emitted (28 Tg), we use equation 12
from Toon et al. (1997) to obtain an area of $4.1x10^4$ km$^2$ for the expected area exposed to
high thermal radiation density from the fireball for a 1 km diameter impactor with an
assumed energy of $6.8x10^4$ Mt. We then multiply that area by 3% (the fraction of C in the
burned fuel that is converted to smoke) and by 2.25 g C cm$^{-2}$, (the assumed carbon
content per unit area of the dry biomass that burns). The user of Table 3 can choose
alternate values of the injected soot by scaling linearly to the biomass concentration they
chose.
Ivany and Salawitch (1993) suggest that the land average above ground biomass was about
$1x10^{18}$ g (about 0.7 g C cm$^{-2}$) at the end of the Cretaceous. The current land average above
ground biomass is about 0.3 to 0.44 g C cm$^{-2}$ (Ciais et al., 2013). An additional 1 to 1.6 g C
cm$^{-2}$ is currently present in the soil, while Ivany and Salawitch suggest 1 g C cm$^{-2}$ in the soil
in the Cretaceous. Some of the soil biomass may burn in a mass fire. Tropical and boreal
forests currently have average biomass concentrations (above ground and in soil) of about
2.4 g C cm$^{-2}$, while temperate forests have about 1.6 g C cm$^{-2}$ including soil carbon (Pan et
al., 2011). Soil carbon is 30% of carbon in tropical forests and 60% in boreal forests.
Together tropical and boreal forests cover 6% of the Earth's surface, and temperate forests



1.5%. These forests cover 26% of Earth's land area. In Table 3 we assume that the
biomass that burns is typical of a tropical or boreal forest assuming the soil carbon burns.
The reader can make other choices for the biomass by scaling from the fuel load that the
reader prefers.
Another modeling issue of concern is the ability of models to follow the initial evolution
of the plume. If we assume that half of the 28 Mt of smoke from the 1 km impact is
injected over an area of $4 \times 10^4$ km$^2$, and over a depth of 6 km near the tropopause as 0.1
$\mu$m radius smoke particles, the smoke will have an initial optical depth near 4000, and the
number density of particles will be about $10^7$ cm$^{-3}$. (The other half of the smoke mass
injected near the ground will likely be removed quickly and have little impact on
climate). Intense solar heating at the top of the smoke cloud near the tropopause will push
it upward, while coagulation will reduce the number of particles by a factor of 2 and
increase their size proportionately in only one minute. Hence, one needs to model this
evolution on sub-minute time scales to accurately follow the initial evolution.
Alternatively, but less accurately, one might spread out the injection in time and space, so
that the climate model can track the evolving smoke cloud using typical model time
steps.

**2. 3 Nano-particles from vaporized impactors**

**2.3.1 Nano-particles from the vaporized material following the Chicxulub impact**

Johnston and Melosh (2012b) calculate that about 44% of the rock vapor that was created
from the K-Pg asteroid impact remained as vapor rather than condensing to form large
spherules in the rising fireball. This vapor is about an equal mixture of impactor and
asteroid, so the 44% mass fraction is approximately equal to the mass of the impactor.
This 44% vapor fraction depends on the pressures reached in the impact, the equation of
state of the materials, as well as the detailed evolution of the debris in the fireball. The
fate of this vapor phase material is not well understood, and has been little studied.
Presently, 100 $\mu$m and larger sized micro-meteoroids ablate to vapor in the upper
atmosphere. Hunten et al. (1980), following earlier suggestions, modeled the
condensation of these rock vapors as they form nm-sized particles in the mesosphere and
stratosphere. Bardeen et al. (2008) produced modern models of their distribution based
on injection calculations from Kalashnikova et al. (2000). Hervig et al. (2006) and Neely
et al. (2011) showed that these tiny particles are observed as they deposit about 40 tons of
very fine grained material on Earth's surface per day. It is likely that a similar process
occurred after the Chicxulub impact. However, in the Chicxulub case the vaporization
occurred during the initial asteroid impact at Chicxulub rather than on reentry of the
material after the fireball rose thousands of km into space and dispersed over the globe.
The presence of 15-25 nm diameter, iron-rich material has been recognized in the fireball
layer at a variety of sites by Wdowiak et al. (2001), Verma et al. (2002), Bhandari et al.
(2002), Ferrow et al. (2011) and Vajda et al. (2015) among others. The nano-phase iron
correlates with iridium, is found worldwide, and therefore is likely a product of the
impact process. Unfortunately, these authors have not quantified the amount of this
material that is present. Berndt et al. (2011) were able to perform very high-resolution





chemical analyses, and also report a component of the platinum group elements that
arrived later than the bulk of the ejecta, and was probably the result of submicron sized
particles. However, they were not able to size the particles, nor quantify their abundance.
In Table 1 we take the injected mass of nano-particles to be 2 x $10^{18}$ g. This choice is
consistent with the vapor mass estimate of Johnston and Melosh (2012b). We assume an
initial diameter of 20 nm, following Wdowiak et al. (2001). We assume the particles are
initially injected over the same altitude range as the Type 2 spherules, because we
speculate that the small particles would not separate from the bulk of the ejecta in the
fireball until the ejecta entered the atmosphere and reached terminal velocity. The mass
injected would lead to an optical depth of particles larger than 1000 even if they
coagulated into the 1 $\mu$m size range. Goldin and Melosh (2009) point out that such an
optically thick layer of small particles left behind by the falling large spheres might also
be important for determining whether the infrared radiation from the atmosphere heated
by the Type 2 spherules is sufficient to start large-scale fires.
The optical properties of the nano-particles are not known. We suggest using the optical
properties of the small, vaporized particles currently entering the atmosphere from Hervig
et al. (2006). These optical constants are plotted in Figure 3. We also assume that the
particles have the density of CM2 asteroids, since Cr isotope ratios suggest that is the
composition of the K-Pg impactor (Trinquier et al., 2006). This density is 2.7 g cm$^{-3}$. A
significant fraction of the vaporized material may be from the impact site, so using an
asteroidal composition to determine the density is an approximation.

**2.3.2 Nano-particles from the vaporized material from a 1 km impact**
Johnson and Melosh (2012b) did not comment on the amount of vapor that would be
expected to not condense as spherules from a 1 km diameter impact. From the theory of
impacts, it is expected that an amount of impactor plus target that is about twice the mass
of the impactor would be converted into vapor from a 1 km diameter impact, just as it is
for a 10 km diameter impact. In Table 3 we assume that about 35% of the impactor mass
plus an equivalent amount of target material, would be left as vapor after spherules form.
We chose this mass fraction, which is lower than that for the K-Pg object, because the
1 km impact will have a smaller fireball, and be more confined by the atmosphere. We
also assume the injected particles will have a diameter of 20 nm. From simple energy
balance along a ballistic trajectory we would expect that the vaporized ejecta in the
fireball from a 1 km impact would rise about a thousand km above the Earth's surface.
This altitude is consistent with limited numerical calculations for large energy releases,
which indicate that the vertical velocity of the fireball is not significantly reduced in
passing through the atmosphere (Jones and Kodis, 1982). As the material reenters the
atmosphere, the particles will come to rest when they encounter an atmospheric mass
comparable to their own mass. Hence it is likely that the altitude distribution of the nano-
particles from the 1km impact will be the same as we have assumed for the K-Pg
impactor in Table 1, which is also similar to, but slightly lower in altitude than the
vertical distribution of micrometeorites on present day Earth as discussed by Bardeen et
al. (2008). It is difficult to determine precisely the area that will be covered by this
material as it reenters the atmosphere. If we assume that it takes about 30 min for the
debris to reach peak altitude and return to the Earth, and that the plume is spreading
horizontally at about 4 km/s then the debris would enter the atmosphere over an area of
about half that of the Earth. These estimates of area covered are consistent with the
observations of the SL-9 impact collisions with Jupiter, and the plume from the much less
energetic impact at Tunguska, though these are not perfect analogs (Boslough and
Crawford, 1997). The optical depth of the nano-particles from the 1 km diameter impact
averaged over the Earth is estimated for comparison with the estimates of other types of
particles to be relatively large, 1.5, as noted in Table 3.

**2.4 Submicron clastics**
**2.4.1 Submicron clastics from the Chicxulub impact**
Another clear component of the K-Pg debris layer is pulverized target material. This
clastic material was first recognized from shocked quartz grains (Bohor, 1990), but there
are also shocked carbonate particles from the Yucatan Peninsula in the K-Pg boundary
layer material (Yancy and Guillemette, 2008; Schulte et al., 2008). Because of chemical
alteration of much of this material in the past 65 million years it is difficult to determine
the mass and size distribution directly except for the shocked quartz, which is readily
identified. The shocked quartz grains generally are large and would not have remained
long in the atmosphere. However, the shocked quartz is probably not directly related to
the bulk of the clastics. For instance, within 4000 km of Chicxulub the shocked quartz is
primarily in the few mm thick fireball layer, which is distinct from the several cm or
thicker ejecta layer that is dominated by clastics. The shocked quartz likely came from
basement rock, reached higher shock pressures than the bulk of the pulverized ejecta and
therefore was distributed globally in the impact fireball along with the melted and
vaporized material from the target and impactor. The other pulverized material, in
contrast, came mainly from the upper portions of the target along with basement rocks
toward the exterior of the crater, and the fragments were distributed locally (within about
4000 km of Chicxulub) in the impact ejecta debris.

The submicron fraction of the clastics is of interest because particles of such size might
remain in the atmosphere for months or years and perturb the climate, unlike larger
particles that would be removed quickly by sedimentation. For instance, Pueschel et al.
(1994) found 3-8 months after the 1991 eruption of Mt. Pinatubo in the Philippines that
volcanic dust particles with a mean diameter near 1.5 $\mu$m were optically important in the
lower stratosphere in the Arctic.
The optical constants for the injected clastics are suggested from their composition. For
the Chicxulub impact the clastic material is largely carbonate evaporates. We suggest
using the optical constants of limestone from Orofino et al. (1998). Unfortunately, the
values need to be generated from a table of oscillator strengths. They also need to be
interpolated into the visible wavelength range. We suggest extending the oscillator
predictions into the visible range as done by Querry et al. (1978). The density of
limestone is in the range of 2.1-2.6 g cm$^{-3}$, while dolomite and anhydrite have densities
near 2.9 g cm$^{-3}$. Granite has a density near 2.6-2.8. While each of these materials
contribute to the clastic debris, for convenience we assume the pulverized ejecta have a





density of 2.7 g cm$^{-3}$.
Pope (2002) and Toon et al. (1997) used two different methods to determine the amount
of the submicron-clastic material from the Chicxulub impact. Unfortunately, these
estimates disagree by about 4 orders of magnitude, as indicated in Table 5, third row,
columns 1 and 2.  Toon et al. (1997) used arguments based mainly on impact models, to
estimate that more than 10% of the mass of the distal layer (> $7x10^{17}$g) is submicron
diameter clastics, which would be significant to climate. Pope (2002) estimated that the
clastics in the distal layer have a mass that is < $10^{14}$g. Pope (2002) used data on shocked
quartz to constrain the amount of clastics, which in principle is a better approach than
using estimates based on a model as in Toon et al. (1997). The amount of clastics of all
sizes in the Pope (2002) model ($10^{16}$g) is only 12-30 times larger than the clastics of all
sizes emitted in the relatively small 1980 Mt. St. Helens eruption. Therefore, based on
Pope's (2002) analysis, the submicron fraction would not be of significance to climate.
Below we attempt to reconcile these two approaches to better determine the amount of
submicron clastics.
**2.4.1.1 Potential errors in the Toon et al. estimate of submicron clastics**
Toon et al. (1997) estimated the amount of submicron **clastics** starting from analytical
models of the mass of material injected into the atmosphere by a 45-degree impact.  They
estimated the mass of melt + vapor per megaton of impact energy  (~0.2 Tg/Mt) and the
mass of pulverized material per megaton of impact energy (about 4.5 Tg/Mt).  Assuming
a $1.5x10^8$ Mt impact, these formulae suggest a melt + vapor amount of $3x10^{19}$g (~$1x10^4$
km$^3$, assuming a density of 2.7 g cm$^{-3}$) and a pulverized amount of $7x10^{20}$ (~$2.5x10^5$
km$^3$).  While sophisticated impact calculations generally agree with the amount of melt +
vapor, not all of it is found to reach high enough velocity to be ejected from the crater.
For example, Artemieva and Morgan (2009) investigated a number of impact scenarios
that created transient craters with diameters of 90-100 km, which they thought to be
consistent with the transient diameter of the Chicxulub crater.  Considering those cases
with oblique impacts from 30-45 degrees with energies of 1.5-2 $x10^8$ Mt, they found that
the melt was in the range $2.6x10^4$ to $3.8x10^4$ km$^3$.  However, the amount that reached high
enough speed to be ejected from the crater was in the range $5x10^3$ to $6x10^3$ km$^3$ (average
$5.6x10^3$ km$^3$, $1.4x10^{19}$g, about 2-10 impactor masses). On average, only about twenty
percent of the melt and vapor amount escapes from the crater. Therefore, Toon et al.
(1997) may have overestimated the amount of melt escaping from the crater by about a
factor of 2.  It should be noted that in Artemieva and Morgan (2009) the melt exceeds the
mass of the distal layer, which is about $4x10^{18}$g, by about a factor of 5, because much of
the melt is deposited as part of the ejecta curtain and never reaches the distal region.
Artemieva and Morgan (2009) find that the total mass ejected from the crater is $1.3x10^4$
km$^3$ ($2.9x10^{19}$ g).  Assuming that 90% of this material is pulverized rock their results
imply that Toon et al. (2007) overestimated the amount of clastic debris ejected from the
crater by a factor of about 25.  In column 3 of Table 5 we correct the amount of
pulverized material to agree with the Artemieva and Morgan (2009) value of $2.9x10^{19}$ g
of clastics escaping the crater.  It is interesting to note that the clastic mass from
Chicxulub is only a factor of about 10 larger than the minimal estimated mass of clastics
ejected in the Toba volcanic eruption about 70,000 years ago (Matthews et al., 2012).



Another issue is the fraction of the pulverized debris that is submicron. Toon et al. (1997)
computed the amount of pulverized debris whose diameter is smaller than 1 $\mu$m from size
distributions measured in nuclear debris clouds originating from nuclear tests that were
many orders of magnitude lower in energy than the K-Pg impact, and from impact crater
studies cited by O'Keefe and Ahrens (1982) based on grain size measurements from
craters. Toon et al. (1997) assume that 0.1% of the total clastic material would be
submicron. Pope (2002) cited studies of volcanic clouds to conclude that 1% by mass of
the pulverized material would be submicron.
Rose and Durant (2009) examined the Total Grain Size Distribution (TGSD) from a
number of volcanic eruptions and concluded that the amount of fine ash is related to
increasing explosivity of the event. The TGSD is supposed to represent the size
distribution as the clastics left the crater. Mt. St. Helens is the most likely of the volcanic
eruptions they considered to be relevant to the extreme energy release in a large impact.
About 2% of the total ejecta from Mt. St. Helens had a diameter smaller than 1$\mu$m. Since
the erupted mass was about 3-8x10$^{14}$ g, the submicron mass emitted by Mt. St. Helens
was about 6-16x10$^{12}$g. Matthews et al. (2012) considered the Toba eruption, whose
clastics are within an order of magnitude of those from Chicxulub. Their data shows that
1-2% of the mass of the clastics is in particles smaller than 1 $\mu$m and 2-6% in clastics
smaller than 2.5 $\mu$m.
In Table 5 we use 2% of the pulverized material as a revised estimate for the fraction of
the clastic material that is released as submicron ejecta. This fraction is a factor of 20
larger than the one used in Toon et al. (1997). Hence our revised submicron mass
estimate for the Chicxulub impact (column 3 row 3) is very similar to the one Toon et al.
(2007) estimated (column 2 row 3) because, although we lowered the estimate of the
clastic mass exiting the crater to agree with Artemieva and Morgan (2009), we increased
the estimate of the fraction that is submicron.
A confounding issue is the amount of submicron and other clastics that escapes from the
near crater region and is distributed globally. A large fraction of the pulverized debris in
the ejecta curtain was removed within 4000 km of the impact crater (Bohor and Glass,
1995), and volcanic ejecta is likewise largely removed near the volcanic caldera. For
example, there is 4-8 cm of ash 3000 km from the Toba crater, which is not too different
from the thickness of the Chicxulub deposits at a similar distance from the crater. If the
removal occurred only by individual particle sedimentation, one could simply take the
mass in the smaller ranges of the size distribution and assume it spread to the rest of the
globe. However, it is clear from volcanic eruption data that a significant fraction of the
submicron debris is removed near the volcano by processes other than direct
sedimentation (Durant et al, 2009; Rose and Durant, 2009). These processes include
rainout of material from water that condenses in the volcanic plume, and also
agglomeration possibly enhanced by electrical charges on the particles. It is likewise
clear that such localized removal occurred after the K-Pg impact. Yancy and Guillemette
(2008) describe accretionary particles that make up a large fraction of the debris layer as
far as 2500 km from the Chicxulub crater. These agglomerated particles, which range in
size from tens to hundreds of $\mu$m, are composed mainly of particles with a radius of 1-4
$\mu$m. While largely composed of carbonate, the particles are enriched in sulfur.
One can use the size distributions from volcanic data, along with the total clastic mass



ejected from Chicxulub to compute the particle agglomeration, and thereby follow the particles as they spread across the Earth. Such work is now being done for volcanic events, for example by Folch et al. (2010). They find that they can successfully reproduce mass deposited on the surface from the Mt. St. Helens eruption by including agglomeration. However, such calculations for Chicxulub are difficult for several reasons: the large clastic masses involved exceed the mass of the atmosphere for a considerable distance from the crater, so the debris flows cannot be reproduced in standard climate models; the complexity of the distribution of material in the plume with some material reaching escape velocity and other parts being hurled over a substantial fraction of the planet make it difficult to determine the spatial distribution of the material, and some material is likely lofted well above the tops of most climate models; and the presence of clastics, melt and rock vapor together with sulfur and water produces a chemically complex plume.

Eventually it will be necessary to use detailed non-hydrostatic, multiphase plume models including agglomeration to better understand the distribution of Chicxulub ejecta. In the meantime for climate modeling we suggest placing the clastic mass in Table 5 ($2.9 \times 10^{19}$ g) in a circular area with radius of 4000 km, which is 22.4% of the area of Earth. This will result in a column density of 25 g cm$^{-2}$, or a layer thickness of about 10 cm. The mass density of the atmosphere is about 1000 g cm$^{-2}$, so this is about a 2.5% perturbation to the mass of the atmosphere. In reality the mass is concentrated near the crater as shown by Hildebrand (1993). However, the observed mass density is relatively constant between 1000 and 4000 km. The initial vertical distribution of this material may be very complex due to density flows within several hundred km of the crater. We suggest initializing models assuming an injection with an altitude independent mass mixing ratio of about 2.5%. Given our suggested vertical distribution 90% of the material will initially lie in the troposphere. Tropospheric material is unlikely to become globally distributed even if it escapes agglomeration, because it will quickly be removed by rainfall.

As an alternative to the complexity of modeling the loss of this material in the troposphere and considering the entire size distribution, we suggest simply placing an appropriate mass into the stratosphere. The values for a stratospheric injection are given in the bottom row of Table 5 and the first row of Table 1. For illustration, we have estimated the final optical depth assuming that 10% of the submicron material (the amount placed into the stratosphere) will escape removal. For a size distribution we suggest using the smaller size mode measured in the stratosphere after the Mt. St. Helens eruption as summarized by Turco et al. (1983). This size distribution is log-normal (Eq. 1), with a mode radius of 0.5 $\mu$m and a standard deviation of 1.65. The estimated optical depth of 88 is very large, even though the submicron clastic material in this estimate is only about 1% of the mass of the distal layer.

### 2.4.1.2 Potential errors in the Pope (2000) estimate of submicron clastics

Pope (2002) determined the amount of clastics by modeling the amount of quartz in the distal layer. He found that he needed an initial injection of about $5 \times 10^{15}$ g of quartz to match the distribution of quartz mass with distance from the impact site. It is not clear



how good this estimate is because, as discussed below, the removal rate of material in
large volcanic clouds, a possible impact analog, does not occur by individual particle
sedimentation, but rather by settling of agglomerates. Hence removal in the region near
the impact site may have been larger than Pope estimated, requiring a larger volume of
quartz; or the removal of clastics may be different than that of quartz. The value in
Artemieva and Morgan (2009) for the pulverized material ejected from the crater is 3
orders of magnitude larger than the estimate of Pope (2002). Most of this material is in
the ejecta curtain, not in the impact fireball, and so is deposited close the impact crater.
The shocked quartz is primarily associated with the impact fireball, so the bulk of the
pulverized material may not be seen in Pope's analysis.
Pope assumed that quartz composed 50% of all the clastic debris, so that all of the
clastics injected weighed about $10^{16}$ g. This number is about two orders of magnitude less
than the clastics from the Toba eruption (Matthews et al., 2012), and more than 3 orders
of magnitude less than the Artemieva and Morgan (2009) estimate for clastics from the
Chicxulub impact.
The assumption by Pope (2002) that quartz is 50% of all the clastics is likely in error.
There is no reason to think there is much quartz in the upper layers of sediment at the
Chicxulub site. In the stratigraphic columns shown by Ward et al. (1995) the pre-impact
sediments at Chicxulub consist of approximately 3 km of Mesozoic carbonates and
evaporites with ~3-4% shale and sandstone. Therefore, it is more likely the quartz
originates from the basement rocks. There is also not a strong connection between the
physical processes that distributed the quartz (the impact fireball, with high ejection
velocity), and those that distributed the pulverized material (the ejecta curtain with low
velocity).
It is possible that the quartz to clastics ratio is determined by the ratio of quartz to total
debris in the samples closest to Chicxulub, since these may have suffered the least removal
by sedimentation. Pope suggests these intermediate distance layers contain about 1%
quartz, but only considers the fireball layer, which is less than 10% of the total ejecta layer
within 1000 km of the crater. The remainder of the intermediate distance layer contains
little quartz, so the clastics could be more than 1000 times the mass of the quartz. It is not
clear that 1000 is an upper limit to the ratio of clastics to quartz because the quartz and
pulverized material move along different paths in the debris cloud. If we accept this ratio
of 1000 for the ratio of clastics to quartz, the mass of clastics from Pope's analysis would
be $5 \times 10^{18}$g, which is within a factor of 6 of the Artemieva and Morgan (2009) value. If 1%
of this mass is submicron then $5 \times 10^{16}$ g of submicron clastics would have been injected into
the upper atmosphere.

**2.4.1.3 Reconciliation of Pope (2000) and Toon et al. (1997) estimates of submicron**
**clastics**

Table 5 shows that the new estimate of submicron mass following the procedure of Toon
et al. (1997) agrees with the new estimate following the procedure of Pope (2002). The
new estimate is about 12 times less than the Toon et al. (1997) value mainly because
Toon et al. (1997) did not consider that most of the pulverized mass would not be ejected
from the crater. The new application of the Pope (2002) approach is about 500 times
larger than the one used by Pope (2002), mainly because we have assumed the ratio of



quartz to clastics is about 1000, rather than 1 as assumed by Pope (2002). Despite the,
perhaps coincidental, agreement of these two estimates, there is substantial uncertainty in
the true mass of submicron clastic particles in the K-Pg distal layer. Observations of the
submicron material in the distal layer are needed.

**2.4.2 Submicron pulverized rock from a 1 km diameter impactor**
In order to determine the properties of the pulverized ejecta from a 1 km impactor, we
use the pulverized mass injection per Tg of impact energy from Toon et al. (1997), but
reduce it by the factor of 25 discussed earlier to account for the fraction of the clastic
mass with enough velocity to escape the crater. This procedure yields a clastic mass of
$1.3 \times 10^{16}$g. For reference, the volume of clastics from the eruption of Mt. Tambora in
1815 is estimated to have been about 150 km$^3$, which is a mass of about $3 \times 10^{17}$ g. Hence
the Tambora eruption likely surpasses the clastics from the hypothetical 1 km diameter
impactor by more than a factor of 10. The same size distribution for the clastics is
recommended for the 1 km impact and the Chicxulub impact, since it seems to hold for a
range of volcanic events from Mt. St. Helens to Toba, which span the 1 km diameter
impactor in terms of clastics. We also suggest that the mass be initially mixed uniformly
in the vertical above the tropopause. According to Stothers (1984) the Tambora clastics
were deposited in layers that are centimeters in thickness at distances 500 km from the
volcano. Accounting for the drift of the ash downwind, the area of significant ash fall
was about $4.5 \times 10^5$ km$^2$. If this same area is used for the initial injection of the clastics for
the 1 km impact, then the column mass concentration is about 8.7 g cm$^{-2}$, which in turn is
slightly less than 1% of the atmospheric column mass. The estimated optical depth of the
clastics in Table 3 is about 25% of the optical depth from nano-particles originating from
vaporized rock. Given that these materials are much less absorbing than soot, and lower
in optical depth than nano-particles they can probably be neglected in estimates of the
climate changes due to a 1 km diameter impact on land.
**3. Gas injections**
There are a large number of gases that might be injected into the atmosphere after an
impact and might be important to atmospheric chemistry, climate, or both. These can
originate from the impactor itself, from ocean or ground water, or from the target
sediments. They may also originate in response to environmental perturbations, such as
wildfires, or atmospheric heating from the impact fireball and ejecta. Various estimates
have been made for each of these sources. However, clear evidence from the distal layer
is not available for any gases of potential interest. Some gases, such as carbon dioxide,
would have stayed in the gas phase rather than condensing into particulate form. Other
gases, such as those containing sulfur, may have reacted on the particles composing the
distal layer, or formed independent particles. In either case sulfur is so common in the
environment it is difficult to detect an injection. For these reasons all the gas phase
injections are uncertain. Below, we first discuss the chemical content of each of the
potential sources of gases, and then we discuss the likely amounts of each material
injected following an impact. Relevant ambient abundances are given in Table 2 and 4
along with estimated injections for Chicxulub and 1 km impacts. The ambient masses are
given to assist the reader in understanding the magnitudes of the injections. Generally
ambient concentrations are given in the literature in terms of the mixing ratio. To
compute the masses we assume the ambient mixing ratios are constant over the whole
atmosphere, or the stratosphere. We then convert the volume mixing ratio to the mass
mixing ratio using the molecular weight and then multiply by the mass of the atmosphere
above either the surface, or tropopause to obtain the total mass of the gas. The ambient
abundances assume the current stratospheric mixing ratio of Cl is 3.7 ppbv (Nassar et al.,
2006), Br is 21.5 pptv (Dorf et al., 2006), inorganic I is 0.1pptv (upper limit from Bosch
et al. 2003), $CO_2$ is about 395 ppmv, and methane is about 1.8 ppbv. Stratospheric S,
taken from the Pinatubo volcanic eruption, is about 10 Tg (Guo et al., 2004), Reactive
nitrogen, $NO_x$, in the stratosphere is difficult to quantify simply. Instead we compare
with the ambient abundance of $N_2O$ in the stratosphere, about $2 \times 10^{14}$ g N. $N_2O$ is a major
source of $NO_x$.

### 895 3.1 Impactor

### 896 3.1.1 Composition of the impactor

Kring et al. (1996) summarized the S, C, and water contents of a large number of types of
asteroids. Trinquier et al. (2006) found from chromium isotopes that the Chicxulub
impactor was most likely a carbonaceous chondrite of CM2 type. Such asteroids have
3.1wt % S, 1.98 wt% C, 11.9 wt% water, and a density of 2.71 g cm$^{-3}$. Over the range of
chondrites, which constitute 85% of meteorite falls, S varies from 1.57 to 5.67 wt%, C
from 0.04 to 3.2 wt %, and water from 0.2 to 16.9 wt %. Kallemeyn and Watson (1981)
report that by mass CM carbonaceous chondrites contain about 4ppm Br. Goles et al
(1967) report that Cl ranges from 190-840 ppmm of carbonaceous chondrites, Br ranges
from 0.25 to 5.1 ppmm, and Iodine ranges from 170 to 480 ppbm. Table 6 summarizes
the composition of asteroids using values for CM2 type carbonaceous chondrites from
Kring et al. (1996) for S, C, and water, and for the Mighei (the CM2 type example) from
Goles et al. (1967) for Cl, Br and I.

### 909 3.1.2 Gases from the impactor

Tables 2 and 4 indicate the direct contributions from 1 and 10 km impactors of a number
of chemicals, as discussed further below. We assume that the entire 10 km or 1 km
diameter impactor melted or vaporized so that all of the gases are released. For the 10
km impactor these gases would have been distributed globally in the hot plume along
with the melt spherules within hours. They would reenter with the same vertical
distribution as the Type 2 spherules. For the 1 km diameter impactor, the initial injection
may have only covered half the Earth, with global distribution over days via wind, after
reentry into the upper atmosphere.
We further assume that the vapors under consideration do not react with the hot mineral
grains either in the plume or in the hot layer at the reentry site. In fact, given the large
particle surface areas in the atmosphere over the globe it is possible that there was a
significant transfer of material from the gas phase to the surfaces of the mineral grains in
a short period of time.
As pointed out by Kring et al. (1996) and Toon et al. (1997) the S in a 10 km diameter
impactor would exceed that from the Mt. Pinatubo volcanic injection by a factor above



1000. Even a 1 km diameter carbonaceous chondrite could deliver several times as much
sulfur to the atmosphere as did the Mt. Pinatubo eruption in 1991. Stratospheric water
could be enhanced by a factor of more than 100 from the water in a 10 km impactor. Cl
could be enhanced by factors above 500, Br by almost 500, and I by more than 50,000.
However, there is not enough C in a 10 km asteroid to affect the global carbon cycle
significantly.
Many investigators have pointed to sulfate as an important aerosol following the
Chicxulub impact. Tables 1 and 3 compare the mass of sulfur from the impactor with the
mass of the spherules and nano-particles. The optical depth, which controls the climate
change following the impact, and the particle surface area, which likely controls
chemistry, are approximately linear with the mass. In our estimates, the sulfate coming
directly from the asteroid could have a large optical depth assuming it was not removed
on the spherules, or large clastics.

**3.2 Seawater**
**3.2.1 Composition and depth of seawater**
The composition of seawater is given in Table 6 (Millero et al., 2008). It is thought that
injections of water into the upper atmosphere will lead to droplet evaporation, with small
crystals of salt left behind. If liquid water is left after a massive injection of water, the
droplets will likely freeze leaving salt behind as particles embedded in ice crystals.
Vaporization of water during the impact may leave behind salt crystals, or the salts may
decompose into their components. As discussed by Birks et al. (2007), complex
simulations are needed to determine how much material is freed from the salt particles to
enter the gas phase where it might destroy ozone. In Table 2 and 4 we list the total
amounts of several interesting chemicals that might be inserted into the stratosphere.
However, all of them except water vapor are likely to be in the form of a particulate until
photochemical reactions liberate them.
A significant uncertainty related to any oceanic contribution to atmospheric composition
is the depth of the ocean in relation to the size of the impactor, and the water content of
sediments at the crater site. The depth of the ocean at the time of the impact is not known.
Many investigators have referred to it as a shallow sea. However, Gulick et al. (2008)
estimates that the water depth averaged over the impact site was 650 m, which is
considerably deeper than earlier estimates. We use a water depth of 650 m in Table 2 to
estimate the amounts of material injected by Chicxulub. A 1 km diameter impactor is
smaller than the average depth of the world oceans, which is about 3.7 km.
**3.2.2 Gases from Seawater-Chicxulub**
For the Chicxulub impact, Pope (1997) assumed that the 650 m depth of seawater within
the diameter of the impactor (10 km) will be vaporized, follow the path of the Type 2
spherules, and reenter the atmosphere globally. In Table 2 we compute the water
vaporized following the equations in Toon et al. (1997). These equations, assuming an
impact velocity of 20 km s$^{-1}$, led to an order of magnitude greater injection of water than
using Pope's estimate. The vaporized water is 0.4 times the impactor mass. During the
vaporization of the seawater we assume the water will be present as water vapor, and that



the materials in the water will be released as vapors. Some of these materials likely
would react quickly with the hot minerals in the fireball or later with the hot minerals in
the reentry layer.
It is also likely that a considerable amount of water was splashed into the upper
atmosphere. Ahrens and O'Keefe (1983) estimated that the water splashed above the
tropopause from a 10 km diameter impact into a 5 km deep ocean would be 30 times the
mass of the impactor. We assume that the amount of water splashed above the tropopause
will scale linearly with the depth of the ocean. Therefore, about 4 times the impactor
mass of water may have been splashed into the upper atmosphere. Much of this water
may immediately condense and rainout, as discussed in Toon et al. (1997). However,
some of the dissolved salts may be released if some of the water evaporates. The
assumed injection of gases, and particulates that might become gases, from the ocean is
summarized in Table 2 for the Chicxulub impact.

### 3.2.3 Gases from Seawater-1 km asteroid

No seawater is injected by the 1 km diameter asteroid impact on land. If a comet hit the
land there would be a water injection.
Pierazzo et al. (2010) estimated that 43 Tg of water would be injected above 15 km by a
1 km asteroid impact into the deep ocean. Of this water, 25% is in the form of vapor and
75% in the form of liquid water. In their modeling the water was assumed to be
distributed with a uniform mixing ratio from the tropopause to the model top. It was also
spread uniformly over an area 6200x6200 km in latitude and longitude. Using the
equations in Toon et al. (1997) for the vaporized water produces a value which is 60% of
the vaporized water from the detailed modeling used in Pierazzo et al. (2010). Given
these water injections we use the composition of sea water to determine the injections of
the various species. Pierazzo et al. (2010) estimate injections of Cl and Br that are more
than an order of magnitude smaller than ours because they consider the amounts that have
been converted into gas phase Cl and Br by photochemical reactions in the atmosphere,
while we estimate the total injections, which initially are likely to be in the particulate
phase.

### 3.3 Impact Site

### 3.3.1 Composition of the impact site

The sea floor at the Chicxulub impact site, like the modern Yucatan, contained abundant
carbonate and sulfate rich deposits. Ward et al. (1995) conclude that 2.5-3 km of
sedimentary rock were present at Chicxulub, composed of 35-40% dolomite, 25-30%
limestone, 25-30% anhydrite, and 3-4% sandstone and shale. The dolomite and
limestone are no doubt porous. Pope et al. (1997) estimate the carbonates in the Yucatan
have a porosity of 20%. The pores would have been filled by seawater since the
sediments were submerged. This ground water produces an equivalent water depth of
about 400 m. The carbon content of limestone is 12% by weight, and of dolomite 15%
by weight. The sulfur content of anhydrite is 23.5% by weight. To our knowledge, trace
species such as Br, Cl, and I have not been reported for these sedimentary rocks, but
would be present in the seawater in the pores.



### 3.3.2 Gases from the impact site

For the 10 km Chicxulub impact we follow Pope et al. (1997) for the abundances of S and C assuming 30% anhydrite, 30% limestone and 40% dolomite. The composition of the impact site is given in Table 6. We ignored species other than S and C that might be in the target material. It is difficult to follow the target debris since some of it is vaporized, and some melted. We follow Pope (1997) and assume that the upper 3 km of the target is vaporized within the diameter of the impactor. The gases within this volume of vaporized material are assumed to be released, and to follow the trajectories of the Type 2 spherules. Pope et al. (1997) estimated the amount of material that would be degassed from target material that was melted or crushed in a large impact. We use the values from Table 3 of Pope et al. (1997) for out of footprint vapors, in our Table 2 for the degassed impact site emissions. We also assume that the granite underlying the impact site does not contribute.

The source gases from a 1 km land impact would depend on the composition of the impact site, so we do not list values in Table 4. We assume nothing would be liberated from the sea floor in a 1 km impact in the deep ocean.

### 3.4 Fires

### 3.4.1 Composition of Smoke

It is well known that forest fires emit a wide variety of vapors into the atmosphere. Andreae and Merlet (2001) provide emission ratios (g of material emitted per g of dry biomass burned) for many vapors expected to be important in the atmosphere as listed in Table 6. As discussed in section 2.2.1, the soot emission may have been enhanced relative to wildfire estimates by Andreae and Merlet (2001) after the Chicxulub impact because the impact-generated fires were mass fires. We do not consider any enhancements of the gas phase emission ratios, but they may also be impacted by fire intensity or the types of plants making up the biomass.

### 3.4.2 Gases from Fires

In Tables 2 and 4 we computed the burned mass from Chicxulub assuming that 1.5 g cm$^{-2}$ of dry biomass burns over the entire land surface area of the Earth, and then used the emission factors from Andrea and Merlet (2001) to obtain the gas phase emissions. For a 1 km impact we assume the area burned is $4.1 \times 10^4$ km$^2$ (Toon et al., 1997), and the dry biomass is 2.25 g C cm$^{-2}$. We then used the emission ratios from Andreae and Merlet (2001) to compute the gas phase emissions. Comparing the gas phase emissions from fires in Tables 2 and 4 with ambient values indicates that there would be large perturbations for all gases for the 10 km diameter impact. Only iodine is significantly perturbed for the 1 km impact. For the gas phase emissions we suggest using the same vertical profile as suggested for soot earlier. The emissions would only occur over the region near the impact site for the 1 km impact.

### 3.5 Gases generated by atmospheric heating



The energy deposited in the upper atmosphere by the initial entry of the bolide, as well as
by the rising fireball, may have converted some $N_2$ to NOx.  Early studies suggested that
a large fraction of the impact energy would be put into the lower atmosphere, which in
turn led to suggestions that a large amount of nitrogen oxides would be produced from
the heated air.  However, it is now understood that most of the energy release from an
impact to the atmosphere will occur at high altitude from reentry of spherules and other
debris.  Toon et al. (1997) reviewed the various ways in which NOx might be generated
following an impact, largely following Zahnle (1990).  They concluded that $3 \times 10^{16}$ g of
NO might be produced from the atmosphere for a 10 km diameter impact with about half
coming from the plume at the impact site, and half from the reentry of material across the
Earth. We have recorded this value in Table 2. For comparison, Parkos et al. (2015)
conducted detailed evaluations of the NOx produced by the infalling spherules and
concluded the spherules could produce $1.5 \times 10^{14}$ moles of NOx ($3 \times 10^{15}$g if the NOx is in
the form of NO) which they further concluded was not sufficient to acidify ocean surface
waters. In Table 2 we use the Toon et al. (1997) injection of NO since it includes both
source mechanisms.  From Zahnle et al. (1990) a 1 km land impact might produce $0.6 \times$
$10^{14}$ g of NO, largely in the hot plume at the impact site.  This value is entered in Table 4.
For comparison, we note that Pierazzo et al. (2010) suggested that the mass of NO
produced by a 1 km ocean impact is about $0.39 \times 10^{14}$ g.

**3.6 Discussion of gas injections**
Some of the gas phase sources just discussed are easy to apply to an impact.  For
example, the emissions from fires simply depend on the area burned, the fuel loading and
the emission factors.
Other sources of gases are more difficult to evaluate. Since we have no measurements for
large impacts, the form of emission can be uncertain. For example, sulfur could be
injected as $SO_2$ or $SO_3$.  Another difficulty that comes in understanding the contribution
of target material to gases, such as $SO_2$, is the pressure needed to vaporize the material.
Pope (1997), for example, adopted pressures above 70 GPa to vaporize carbonate, 100
GPa for complete vaporization of anhydrite, and 10 GPa for water vaporization from
pores. These vaporization pressures are higher than suggested by early researchers,
leading to lower amounts of target vaporized.  Pierazzo et al. (2003) redid the impact
calculations and also estimated the amounts of materials that might be released, which are
close to those estimated by Pope (1997).  The altitude distribution of the ejecta varies
with the source of the material.  Finally the chemical form of the emission varies with
thermochemistry in the ejecta plume or fireball, and interactions with hot mineral
surfaces, and for some materials exposure to high temperature on reentry.
Tables 2 and 4 summarize our choices for the injections of the various gases.  For each
type of source we also specify the altitude of the expected injection, using a reference to
Table 1 and 2 for the particle injections.  We assume all of the impactor mass entered the
rising fireball, so it would be injected near 60 km altitude along with the spherules.  In
some cases, for example for the degassed target material and for splashed seawater, we
consider the material to have been uniformly mixed above the tropopause.  For materials
coming from fires we assume the same vertical injection as for soot.



As has been pointed out many times (Kring, 1996; Toon et al., 1997; Pope, 1997;
Pierazzo et al., 2003) the sulfur injection from a 10 km impactor might be thousands of
times greater than that from the Pinatubo eruption, and also was likely larger than the
injection from the massive Toba eruption by a factor between 10 and 100. Our sulfur
injection from the target material is about half that of Pope's (1997) estimate of $10^{17}$ g and
slightly less than Pierazzo et al's (2003) estimate for a 15 km diameter impactor of 7.6 x
$10^{16}$ g. Our sulfur injection from the asteroid itself is within the range suggested by Pope
(1997) of 2.7-5.9 x $10^{16}$ g. Interestingly, the sulfur injection we estimate for Chicxulub is
about 10 times greater than the yearly emission estimated by Schmidt et al. (2015) for a
large flood basalt from the Deccan traps. Of course, the flood basalt might continue for a
decade or more, bringing the total sulfur emission close to that from the Chicxulub
impact. Table 4 suggests that the sulfur injection from a 1 km impact would be several
times greater than that from the Pinatubo eruption, but that would be only a modest
injection relative to historical volcanic eruptions. In Table 1 and Table 3 we assume the
injected sulfur gas is converted into sulfate. If so it would yield a large optical depth for
the Chicxulub impact. However, for both the 1 km and Chicxulub impacts, the sulfur
injection, if converted to sulfate, would be an order or magnitude less massive than the
nano-particles. Therefore, the sulfate would be an order of magnitude less important
optically than the nano-particles. While it might exceed the soot mass slightly, soot is
much more important optically than sulfate, which is transparent at visible wavelengths.
Therefore, the sulfate in our model is of relatively little importance optically, unless the
sulfur remains in the air after the other particles are removed.
Our estimated C injection (in the form of $CO_2$) is dominated by emissions from forest
fires. We have the same emission from the impactor as Pope (1997), but we have less
than half the emission from the target material as Pope (1997) or Pierazzo et al. (2003).
All these studies suggest a small impact perturbation relative to the $CO_2$ 65 million years
ago, which was several times larger than now.
The water vapor injections in Tables 2 and 4 are very large compared with ambient
values in the stratosphere. However, most of the water is from fires, and half will be
injected into the troposphere where it will be quickly removed. The water from the
impactor and target is modest, about 1 cm as a global average depth of rain. The typical
rainfall averaged over the current Earth is about 3 mm day$^{-1}$. The emissions from the
impactor and from vaporized seawater, both of which would have been injected globally
at the same altitudes as the Type 2 spherules, are capable of saturating the entire ambient
stratosphere. Our water injection is similar to that estimated by Pope (1997), and Pierazzo
et al. (2003). While the water vapor has been largely ignored in previous work on the
Chicxulub impact, it has the ability to alter the thermal balance of the stratosphere by
emitting and absorbing infrared light. Water vapor may have been a factor in the
radiation of thermal energy to the surface during the first few hours after the K-Pg
impact, since Goldin and Melosh (2009) sought an infrared absorber to prevent radiation
from escaping from the top of the atmosphere. Some of the particles in the stratosphere
might be removed by precipitation, but the mass of water injected is comparable to the
mass of the nano-particles and spherules. Therefore, removal by precipitation is probably
not significant since if the water condenses on all the particles it will add only a small
mass, and increase the fall rate only slightly, while if water condenses on only a subset of





the particles it will remove only a subset. The water injection by the 1 km diameter
impact on land is about 15% of the ambient water, but might still lead to some significant
perturbations if it is injected into the upper stratosphere. The 1 km impact in the deep
ocean could inject about 40 times the ambient water into the stratosphere (Pierazzo et al.,
2010), and water should be considered in simulations of such impacts.
For the 10 km diameter impactor, there are injections of Cl, Br, and I that exceed the
ambient values by orders of magnitude. There are significant sources for all three
halogens from fires, the impactor and seawater, so it seems inescapable that large
injections would have occurred. The injections of $NO_x$ from fires, and from heating the
atmosphere are also very large compared with ambient values. For instance, Table 2
shows the $NO_x$ injections are one to two orders of magnitude larger than the stratospheric
burden of $N_2O$, the principle source of NOx. For the 1 km diameter land impact only the
injections of I and $NO_x$ appear large enough to perturb the chemistry of the stratosphere.
However, as discussed by Pierazzo et al. (2010) significant Cl and Br injections could
occur for a 1 km impact in the ocean. Seawater injections of Cl, Br, I, and S are
complicated because the salts may be injected in particulate form.

## 4. Implications for climate, atmospheric chemistry and numerical modeling, and suggestions for future data analysis

Since the discovery of the K-Pg impact by Alvarez et al. (1980), many papers have
speculated on which of the many possible effects of the impact on the environment could
have caused the mass extinction. It has become fashionable to claim that one or another
effect is dominant. However, it is quite likely that several effects overlapped, each of
which might have been devastating to a particular species or ecosystem, but which
together made survival very difficult for a broad range of species distributed over the
globe. Here we summarize the environmental perturbations we find likely. However,
there are many uncertainties, and additional data are needed. We outline the data that
would be useful to obtain from the geologic record, and summarize it in Table 7. Also,
models have barely scratched the surface of what is possible in better understanding of
the post impact environment. We summarize the types of modeling work that would be
interesting to pursue. We extend these ideas to smaller impacts since more than 50
impacts of kilometer-sized objects may have occurred since the extinction of the
dinosaurs.
Table 1, shows that spherules, soot, nano-particles, submicron clastics, and sulfates each
may have had very large optical depths. An optical depth greater than unity could have
serious consequences for the environment if maintained for very long. Each of these
materials was likely present in the atmosphere, so they may have interacted.
The spherules are unlikely to have changed climate directly because they would have
been removed quickly from the atmosphere by sedimentation due to their large size.
However, these particles, together with the other impact debris with significant mass,
likely heated the upper atmosphere to temperatures between 1000 and 2000K. The high
temperature upper atmosphere would then have irradiated the surface with near infrared
radiation, causing forest fires. Wolbach et al. (1985) first recognized that the global biota
likely burned after the impact, and Melosh et al. (1990) identified the mechanism for



starting the fires. The recent work by Goldin and Melosh (2009) identified some
complexities in the ignition mechanisms that need further work to be understood. They
pointed out that the light might be blocked by the large spherules falling below the heated
atmospheric layer. However, this is a complex problem since water vapor, and vaporized
impactor would have been present to block radiation escaping to space. Also convection
should occur in such a strongly heated layer, which would act to retard the fall of the
particles as it does for hailstones in tropospheric convection. These issues all deserve
further study with suitable models. Furthermore, evidence for the nano-particles should
be sought as discussed further below.
Robertson et al. (2004) argued that large dinosaurs and other unsheltered animals could
have been killed immediately by the radiation from the sky and the subsequent fires.
However, it is possible there were refugia on the land, either in regions where spherules
did not reenter the atmosphere, as suggested by Kring and Durda (2002) as well as
Morgan et al. (2013), or in regions that happened to have heavy cloud cover which may
have blocked the radiation. To better understand the possibility of refugia, more
complete evidence for the global distribution of spherules would help resolve their
possible non-uniform deposition, as suggested in Table 7. It is known that iridium was
perturbed worldwide following the K-Pg impact. Although iridium concentrations are
spatially variable for a number of reasons, they are basically homogenous over the Earth
and do not fall off with distance from the impact site, or at high latitudes. Similar data on
spherules would be useful to determine if the spherules were injected everywhere, or in
special places. Numerical values of the spherule concentrations and size distributions to
augment the values noted by Smit (1999) would also be of value, as noted in Table 7.
Models of the transmission of the light from the hot debris layer above 60 km through
dense water clouds and the response of the clouds to the heating would be also useful. It
has long been recognized that intense thermal radiation and fires could not have been the
only extinction mechanisms at work, since the mass extinctions in the oceans could not
have occurred in this way, but instead were likely due to the low light levels preventing
photosynthesis (Milne and McKay, 1982; Toon et al., 1982; Pollack et al., 1983; Toon et
al., 1996; Robertson, et al., 2013b). The low light levels would have been caused by the
high optical depths of the soot and nano-particles that remained suspended in the air for a
year or more after the impact.
We know from the work of Wolbach et al. (1985; 1988; 1990; 2003) that there is
abundant soot in the K-Pg distal layer. It is highly likely that the soot originated from
wildfires (Robertson et al., 2013a), but its origin is of secondary concern for climate. The
widespread distribution of the soot in the layer, and the small size of the particles indicate
this material was almost certainly global in extent. Wolbach et al. (1988) show that soot
and iridium are tightly correlated across the K-Pg distal layer. The soot and iridium in
the distal layer must have been deposited within a few years of the impact, since small
particles will not stay in the air much longer. Therefore, any fires must have been within
a year or two of the impact. As noted in Table 7, further examination of the distributions
of soot, iridium and spherules might clarify how long these materials remained in the
atmosphere, which is expected to be days for the spherules, and a few years for the soot
and iridium on small particles. Once in the water column, spherules would fall to the
bottom in days or weeks. However, in the absence of fecal pellets formed by plankton



around the soot, it would take decades for soot to reach the ocean depths by falling.
Currents would likely carry the soot down rather than gravity.
The amount of soot in the K-Pg distal layer would produce a very high optical depth
when it was in the atmosphere. The transmission of light depends not only on the optical
depth, but also on the single scattering albedo of the particles. The single scattering
albedo measures the fraction of the light that is scattered, or absorbed. Scattering light,
which occurs from sulfates that absorb sunlight only weakly, is not nearly as effective in
changing climate as absorbing light.
As discussed by Toon et al. (1997), soot with an optical depth of 100 would prevent any
sunlight from reaching the surface—it would be pitch black. No climate simulations of
such large soot optical depths have ever been conducted. However, there have been
simulations for optical depths in the range of 0.05-1, which show temperatures dropping
to ice age conditions within days, precipitation falling to 50% of normal, and the ozone
layer being destroyed as discussed further below (Robock et al., 2007a,b; Mills et al.,
2008, 2014). There are a number of complexities inherent in climate calculations for soot.
For example, it is important to know how long the soot remained in the atmosphere in
order to determine how long photosynthesis may have been retarded in the oceans. The
lifetime of the soot in turn may depend on the size of the soot particles, their shape, the
amount of rainfall in the lower atmosphere, and the amount of sunlight reaching the soot.
The amount of sunlight reaching the soot matters because heating the soot also heats the
surrounding air, causing it to rise and loft the soot to high altitudes, where it is protected
from rainout (Malone et al. 1985; Robock et al. 2007a,b). These issues can be considered
in modern climate models.
Much of the vaporized impactor and target material is thought to have re-condensed to
250 $\mu$m-sized spherules (O'Keefe and Ahrens, 1982; Johnson and Melosh, 2012b), which
are observed, but a significant fraction may have remained as nanometer sized grains
(Johnson and Melosh, 2012b). Iron-rich, nano-phase material with a diameter of 15-25
nm has been identified in the fireball layer at a variety of sites by Wdowiak et al. (2001),
Verma et al., (2002), Bhandari et al. (2002), Ferrow et al. (2011) and Vajda et al. (2015)
among others. However, the abundance of this nano-phase material is not yet constrained
by observations. As noted in Table 7, it is important to quantify the abundance of this
nano-phase material, and to confirm that it is the remnant of the vaporized target and
impactor. If the estimate of Johnson and Melosh (2012b) of its abundance is roughly
correct then, given the optical depth estimate in Table 1 and its input location in the upper
atmosphere above the soot generated by forest fires, this nano-phase material would be
the dominant source of opacity for changing the climate, and would also greatly affect the
amount of radiation emitted to the surface that could start wildfires in the hours following
the impact. The material contains iron, so it is likely to have been a good absorber of
sunlight. Alternatively, this material might have attached itself to the large spheres and
been quickly removed, though this seems unlikely since the large spheres would separate
gravitationally from the smaller material within hours. No one has yet considered the
effect of this nano-phase material, which is distinct from the clastics envisioned by Toon
et al. (1997) and Pope (2002), on the environment after the K-Pg impact.
The most massive part of the ejecta from the K-Pg crater consisted of clastics: crushed
and pulverized material. Much of this material fell relatively close to the crater, though



significant amounts were emplaced as far a 4000 km from Chicxulub. For comparison the
Toba volcanic eruption about 70,000 years ago is estimated to have released more than
$2 \times 10^{18}$g of clastics (Matthews et al., 2012), a factor of about 15 less than our estimate for
the Chicxulub impact in Table 1, but more than 200 times greater than the upper limit
previously estimate by Pope (1997) for the clastics generated by Chicxulub.
The Toba eruption may have had a significant impact on the climate, as discussed further
below; however, the magnitude of the effect is controversial. Alvarez et al. (1980), as
well as Toon et al. (1982) and Pollack et al. (1983), thought that the K-Pg layer was
dominated by submicron clastics that caused major loss of sunlight at the surface and
consequently very low temperatures. However, while we don't know the fraction of the
layer composed of submicron clastics, it is clear that the layer is both thinner than thought
in the years just after its discovery and also dominated by other parts of the impact debris
such as the spherules and the nano-particles. It would be very useful to measure the
amount of submicron clastics in the K-Pg distal layer. Possibly, as suggested in Table 7,
one could start by identifying the amount of submicron quartz in the layer by searching
for small shocked quartz grains. Toon et al. (1997), and Pope (2002) used two differing
indirect approaches to quantify the submicron clastics, and came up with answers that
differ by a factor of about $10^4$. Here we attempted to reconcile these approaches, with the
result shown in Table 1 yielding a significant optical depth. Although the submicron
clastics by themselves would have produced extreme climate changes if they were as
abundant as we estimate, they would have been less important than the soot, and the
nano-particles given our estimates here. The submicron clastics may have been injected
higher than the soot, but lower than the nano-particles on average. Climate calculations
involving all these materials are needed to understand how they may have interacted in
the atmosphere.
The final particulates with large optical depths in Table 1 are sulfates. Pope et al. (1997),
Pierazzo et al. (2003) and others have advocated for the importance of these particles in
recent years. Unfortunately, sulfates in the K-Pg layer have not been traced
unambiguously to the impact, because sulfur is so common in the environment. Possibly
sulfur isotopic studies could distinguish the sulfur in the impactor from sulfur in the
terrestrial environment, but we are not aware of such studies. While there is little doubt
that large amounts of sulfur were present in the target material and in the asteroid, it is
possible that much of it reacted with the hot rock in the impact plume, or the atmospheric
layer heated by re-entering material. Sulfur is present in impact melt spherules and in
carbonaceous clastics, so not all of it was released to the gas phase. Given the large
opacity of the numerous types of particles in the atmosphere, photochemical reactions
would have been inhibited, which would retard the conversion of sulfur dioxide gas into
sulfate particles. It is possible that measurements of the sulfur mass independent
fractionation (MIF) could reveal whether the sulfur quickly reacted with rocks, which
should yield a MIF of zero, of if the sulfur slowly converted to sulfate, which might lead
to MIF not being zero if resolved over the thickness of the distal layer. It is known that a
non-zero MIF can occur following volcanic eruptions due to time dependent movement
of sulfur between changing sulfur reservoirs in the atmosphere (e.g. Pavlov et al., 2005).
It is not clear if $SO_3$ or $SO_2$ was the dominant sulfur bearing gas in the ejecta plume.
However, the gas phase reaction of $SO_3$ and water is not a simple reaction as often




abbreviated in papers about atmospheric sulfur chemistry, but instead involves water
vapor clusters or $SO_3$ adducts. Sulfur dioxide is observed to convert to particulates with
an e-folding time of less than one month for moderate-sized volcanic eruptions such as
the Mt. Pinatubo eruption. Following the K-Pg impact sulfur dioxide or trioxide gas may
have had an extended lifetime in the atmosphere, due to the lack of sunlight to drive
chemical reactions to convert it to sulfates. Clastics and nano-particles and soot, may
have coagulated to large sizes and fallen out over a year or two. Alternatively, the sulfur
gases may have reacted quickly on all the surfaces present, particularly in hot water
present in the hot radiating layer when the ejecta reentered. Pope (1997) and Pierazzo et
al. (2003) have pointed out the possible importance of the extended lifetime of the sulfate
to causing a prolonged period without photosynthesis in the oceans. However, clastics or
soot needed to be present in the sulfate to achieve the loss of sunlight. Recent work on the
Toba eruption (Timmerick et al., 2010) shows that large sulfur injections do not produce
proportionately larger climate perturbations because the climate effects of sulfur
injections are self-limiting, as originally shown by Pinto et al. (1994) and recognized by
Pope (1997) and Pierazzo et al. (2003). Toba probably injected an amount of sulfur
dioxide within an order of magnitude of that from the K-Pg impact. Larger particles have
smaller optical depths, and shorter lifetimes, than smaller particles that result from
smaller $SO_2$ injections. Further work is needed to understand the chemistry of the sulfur
injected by the Chicxulub impact to determine if it was a significant factor in the
extinction event.
Table 2 shows that significant injections of various ozone destroying chemicals such as
$NO_x$, Cl, Br, and I, likely occurred. The effects of these gases need to be considered in
calculations but, given the expected darkness, photochemistry may have ceased until the
atmosphere cleared.
Table 3 suggests that the much smaller mass injections from the impact of a 1 km
diameter asteroid on land may produce optical depths that may still be important.
Climate models are needed to fully evaluate these perturbations. At first glance the
injections seem small. For example, the sulfur injection is only about 4 times larger than
that from the Pinatubo eruption. However, the soot injection is very large. Robock et al.
(2007a) and Mills et al. (2014) examined smoke injections at the tropopause of about one
third the 1 km asteroid injection near the tropopause and found that the ozone layer was
severely damaged, and low enough temperatures resulted to damage crops for a decade
after the injection. Table 4 also indicates significant injections of iodine, which may
further damage the ozone layer.
About 50 1-km impacts might have occurred since the demise of the dinosaurs. Based on
the fraction of Earth covered by water, about 35 of these would be expected to have hit
the oceans, perhaps resulting in large ozone losses as discussed by Pierazzo et al. (2010).
Each of the 15 impacts that occurred on land might have led to significant injections of
nano-particles. Paquay et al. (2008) recognized the osmium signature of two large
impacts in the Late Eocene, which produce the 100 km diameter craters at Popigai and
Chesapeake Bay. The osmium indicates a substantial input of vaporized impactor to the
atmosphere from collisions of asteroids larger than 1 km in diameter. Climate model
simulations are needed to evaluate the climate changes that might have occurred. The
effects could have been variable for a variety of reasons, including variability in the light



absorbing properties of rock from differing objects. To have injected significant amounts
of smoke the impactor would need to hit a tropical forest, or at least a heavily forested
region. About 26% of the world is currently forested; about 6% is in tropical rain forest.
Forested area has greatly declined. Tropical rainforests might have covered as much as
20% of the Earth until recently. Hence, about 3 1-km objects might have hit a tropical
rainforest and injected significant amounts of smoke since the K-Pg event.
In this work we have established a set of initial conditions (Tables 1-4) that may be used
for modeling the climate and air chemistry after the K-Pg impact, or the impact of a 1 km
asteroid. Other authors have considered some of these initial conditions, but some, such
as the nano-particles from the vaporized impactor, have not been previously studied in
the detail needed to fully evaluate their importance. Much more work is needed to obtain
field data to further constrain some of parameters, and to resolve remaining differences of
opinion about some of the values. However, simulations using these initial conditions can
now be conducted with modern models of climate and atmospheric chemistry, which
should shed light on the environmental conditions at the K-Pg boundary and the dangers
posed by future impacts. We recently completed such simulations using the Whole
Atmosphere Community Climate Model (WACCM) at the National Center for
Atmospheric Research.
**Author contributions:** Owen Toon worked to compile the particle and gas emissions.
Charles Bardeen tested them in a climate model to determine if the initial conditions were
specified completely. Rolando Garcia considered the gases that would be important for
atmospheric chemistry.
**Acknowledgements:** We thank Wendy Wolbach for helpful comments about soot. C.
Bardeen and R. Garcia were funded by NASA Exobiology grant #08-EXOB08-0016. The
University of Colorado supported O. B. Toon's work.



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



Table 1: K-Pg injection scenario for impactor mass $\sim 1.4 \times 10^{18}$ g, impact energy $\sim 2.8 \times 10^{23}$ J$=6.8 \times 10^7$ Mt for 20 km/s impact

| Property/ Constituent | Type 2 spherules | Soot | Nano-particles from vaporized rock | Clastics, $<\mu$m distributed globally | S |
|---|---|---|---|---|---|
| Material amount, g, (g cm$^{-2}$) | $2.3 \times 10^{18}$ (0.44) | $7 \times 10^{16}$ ($1.3 \times 10^{-2}$) | $\sim 2 \times 10^{18}$ (0.4) | $<6 \times 10^{16}$ (0.01) | $9 \times 10^{16}$ ($5.4 \times 10^{-2}$ g SO$_4$/cm$^2$) |
| Estimated global optical depth as 1 $\mu$m particles [*] | $\sim 20$ (for 250 $\mu$m particles) | $\sim 100$ | $\sim 2000$ | $\sim 90$ | $\sim 450$ |
| Vertical distribution | 70 km as center of Gaussian distribution with half width of 6.6 km | Eq. 2 | Same as Type 2 spherules | Uniformly mixed vertically above tropopause | Same as Type 2 spherules |
| Optical properties | Not relevant | $n=1.8$ $k=0.67$ | Hervig et al., (2006) | Orofino et al. for limestone | Sulfuric acid |
| Initial Particle size | 250 $\mu$m diameter | Lognormal, modal particle radius 0.11$\mu$m, sigma 1.6; monomers 30-60 nm | 20 nm diameter | Lognormal, modal particle radius 0.5$\mu$m, sigma 1.65 | gas |
| Material density, g cm$^{-3}$ | 2.7 | 1.8 | 2.7 | 2.7 | 1.8 |

[*] Qualitative estimate for comparison purposes only




Table 2: Gas phase emissions (g) from the Chixculub impact

| Sources/ Gases | S $(10^{13})$ | C (as $CO_2^{**}$) $(10^{17})$ | $H_2O$ $(10^{15})$ | Cl $(10^{12})$ | Br $(10^{10})$ | I $(10^{7})$ | N $(10^{14})$ | Vertical distribution |
|---|---|---|---|---|---|---|---|---|
| Ambient values, g | 1[*] | 8.4 | 1.3 strat | 2.3 strat | 3.1 strat | <2.3 strat | 2 as $N_2O$ | |
| Impactor | $4\times10^3$ | 0.3 | 200 | $7\times10^2$ | $5\times10^2$ | $7\times10^4$ | | As Type 2 spherules |
| Forest fires | 40 | 6 | 1500 | 200 | 1000 | $9\times10^5$ | 10 | As soot |
| Vaporized sea water | 60 | small | 600 | $1\times10^4$ | $5\times10^3$ | 40 | - | As Type 2 spherules |
| Splashed sea water[***] | 500 | small | $5\times10^3$ | $1\times10^5$ | $4\times10^4$ | $3\times10^2$ | - | Uniformly mixed above tropopause |
| Impact site (vaporized) | 5000 | 0.6 | 90 | 800 | 400 | 3 | | As Type 2 spherules |
| Impact site (degassed) | 500 | 0.1 | 120 | $2\times10^3$ | $1\times10^3$ | 7 | | Uniformly mixed above tropopause |
| Atmospheric heating | | | | | | | 300 as $NO_x$ created from air | Half uniformly mixed, half as Type 2 spherules |

• [*] Based on Pinatubo eruption
• [**] Mass as given in terms of C, but emission is in the form of $CO_2$.
• [***] S, Cl, Br, I likely injected as particulates




Table 3: 1 km land[*] injection scenario for impactor mass $1.4 \times 10^{15}$ g; impactor energy
$\sim 2.8 \times 10^{20}$ J $= 6.8 \times 10^{4}$ Mt

| Property/ Constituent | Type 2 spherules | Soot[**] | Nano-particles from vaporized rock[***] | Clastics, $<\mu$m distributed globally | S |
|---|---|---|---|---|---|
| Material amount g (g cm$^{-2}$) | $1.4 \times 10^{15}$ ($2.6 \times 10^{-4}$) | $2.8 \times 10^{13}$ ($5.6 \times 10^{-6}$) | $1 \times 10^{15}$ ($2 \times 10^{-4}$) | $2.6 \times 10^{13}$ ($5 \times 10^{-6}$) | $4.4 \times 10^{13}$ ($2.6 \times 10^{-5}$ g SO$_4$ cm$^{-2}$) |
| Estimated global optical depth as 1 $\mu$m particles | 0.2 (as 15 $\mu$m particles) | $4.7 \times 10^{-2}$ | 1.5 | $4 \times 10^{-2}$ | 0.22 |
| Vertical distribution | Table 1 Over 50% of Earth | Table 1 Over $4 \times 10^{4}$ km$^2$ | Table 1 Over 50% of Earth | Uniformly mixed above tropopause, spread over $4 \times 10^{5}$ km$^2$ | Follow nano-particles |
| Optical properties | Not relevant | Table 1 | Table 1 | Depends on impact site | Table 1 |
| Initial particle size ($\mu$m) | 15$\mu$m | Table 1 | 20 nm | Table 1 | |

[*]We assume a 1 km asteroid impact would not penetrate through the 5km average depth
of the ocean. Therefore, none of the materials in this Table would be injected into the
atmosphere for an ocean impact. For the density of all materials follow Table 1.
[**]The material amount assumes an impact into a region where 2.25 g C cm$^{-2}$ flammable
biomass is consumed. The material amount can be scaled linearly for other choices of
available biomass that burns.
[***]We assume about 35% of the impactor and an equivalent mass of target would vaporize
and end up as nano-particles.





Table 4: Gas phase emissions (g) from a 1-km diameter impact

| Sources/ Gases | S $(10^{13})$ | C[*] $(10^{17})$ | $H_2O$ $(10^{15})$ | Cl $(10^{12})$ | Br $(10^{10})$ | I $(10^{7})$ | N $(10^{14})$ | Vertical distribution |
|---|---|---|---|---|---|---|---|---|
| Ambient values, g | 1[**] | 8.4 | 1.3 strat | 2.3 strat | 3.1 strat | <2.3 strat | 2 as $N_2O$[**] | |
| Impactor/ land only | 4.4 | $3 \times 10^{-2}$ | 0.2 | 0.7 | 0.5 | 68 | - | As type 2 spherules |
| Forest fires/land only | $2.7 \times 10^{-2}$ | $4 \times 10^{-3}$ | 0.9 | 0.12 | 0.62 | 560 | $6.9 \times 10^{-3}$ | As soot |
| Vaporized sea water | 0.9 | small | 10 | 200 | 80 | 0.6 | | Uniformly mixed |
| Splashed sea water[***] | 3 | small | 30 | 600 | 200 | 2 | | |
| Atmospheric heating | | | | | | | 0.6 | Uniformly mixed |

[*]in the form of carbon dioxide
[**]based on Pinatubo volcanic eruption
[***]S, Cl , Br, I may be released as particulates.



Table 5: Comparison of Toon et al. (1997) and Pope (2002) estimates of submicron
clastics.

| Method | Quartz based estimate- Pope (2002) | Injected mass-Toon et al. (1997)[*] | Injected mass - revised | Quartz based estimate- revised | 1 km impactor[**] |
|---|---|---|---|---|---|
| Initial clastic debris, g | $<10^{16}$ | $7 \times 10^{20}$ | $2.9 \times 10^{19}$ | $5 \times 10^{18}$ | $1.3 \times 10^{16}$ |
| % clastic $<1$ $\mu$m | $<1$ | 0.1 | 2 | 1 | 2 |
| Submicron clastics, g | $<10^{14}$ | $7 \times 10^{17}$ | $5.8 \times 10^{17}$ | $5 \times 10^{16}$ | $2.6 \times 10^{14}$ |
| Stratospheric submicron surviving initial removal, g | $10^{14}$ | $7 \times 10^{17}$ | $<5.8 \times 10^{16}$ | $5 \times 10^{16}$ | $< 2.6 \times 10^{13}$ |

[*] assuming an impact energy of $1.5 \times 10^{8}$ Mt, and a velocity of 20 km/s.
[**] scaled from Injected Mass Revised using energy scaling assuming an impact energy of
$6.8 \times 10^{4}$ Mt




Table 6: Impactor composition, seawater composition, Yucatan impact site composition
and forest fire emission ratios

| | S | C | $H_2O$ | Cl | Br | I | EC | N |
|---|---|---|---|---|---|---|---|---|
| Carbonaceous Chondrite (g/g impactor) | $3.1 \times 10^{-2}$ | $1.98 \times 10^{-2}$ | $11.9 \times 10^{-2}$ | $4.7 \times 10^{-6}$ | $3.27 \times 10^{-6}$ | $4.8 \times 10^{-7}$ | | |
| Sea water (g/g sea water) | $9.1 \times 10^{-4}$ | $3 \times 10^{-6}$ | 0.965 | $1.9 \times 10^{-2}$ | $8.2 \times 10^{-5}$ | $6.0 \times 10^{-10}$ | - | - |
| Impact site (g/g site) | $7.1 \times 10^{-2}$ | $9.6 \times 10^{-2}$ | 0.07 | | | | | |
| Emission ratios for forest fires g/g of dry biomass burned | $2.9 \times 10^{-4*}$ | $4.3 \times 10^{-1}$ as $CO_2$ $4.4 \times 10^{-2}$ as CO $5.1 \times 10^{-3}$ as $CH_4$ | Highly variable, can equal dry weight | As $CH_3Cl$ $1.4 \times 10^{-5}$ to $1.3 \times 10^{-4}$ | As $CH_3Br$ $6.7 \times 10^{-6}$ | As $CH_3I$ $6.1 \times 10^{-6}$ | $6.6 \times 10^{-4**}$ | $7.5 \times 10^{-4}$ as NO $6 \times 10^{-5}$ as $N_2O$ |

*The mass is given in terms of S, but the emission is in the form of $SO_2$.
** We used 0.03 g/g in Table 3, because forest fires will not produce as much soot as mass
fires.



Table 7 Suggestions for data collection

| Property of interest | Rationale |
|---|---|
| Global distribution of spherules | Some impact models suggest spherules were not distributed globally, limiting area of Earth that might experience fire ignition |
| Number concentration, size of spherules | Current data are incomplete on number and size of spherules |
| Soot distribution | Profile soot/iridium/spherule distribution to determine if fires are contemporaneous with iridium fallout |
| Nano-meter material | Nano-meter material has been detected, but its mass needs to be quantified |
| Clastics | Submicron component not detected. Possibly search for micron/submicron shocked quartz. |
| Sulfur | Use sulfur isotopes to search for extraterrestrial sulfur, sulfur MIF to test for prolonged lifetime |





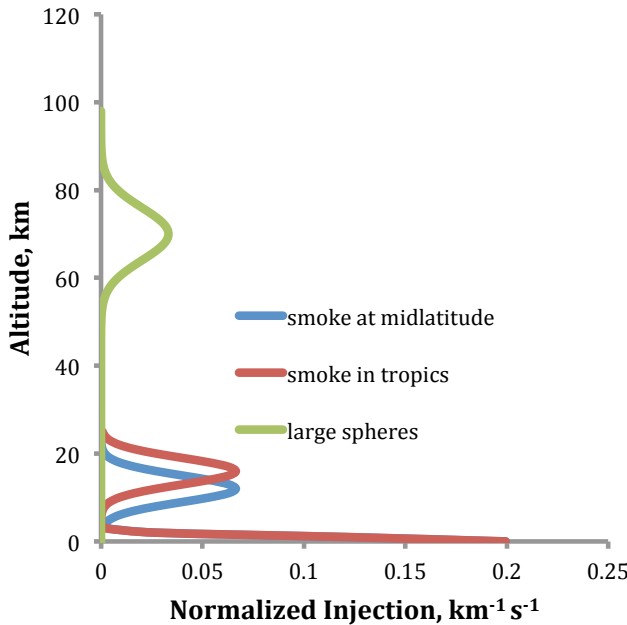

Figure 1. Injection profiles for smoke at midlatitudes and the tropics and for large spherical particles. Many other constituents follow the same vertical profiles as noted in Table 1-4. We suggested clastics be placed above the tropopause using a constant mixing ratio.





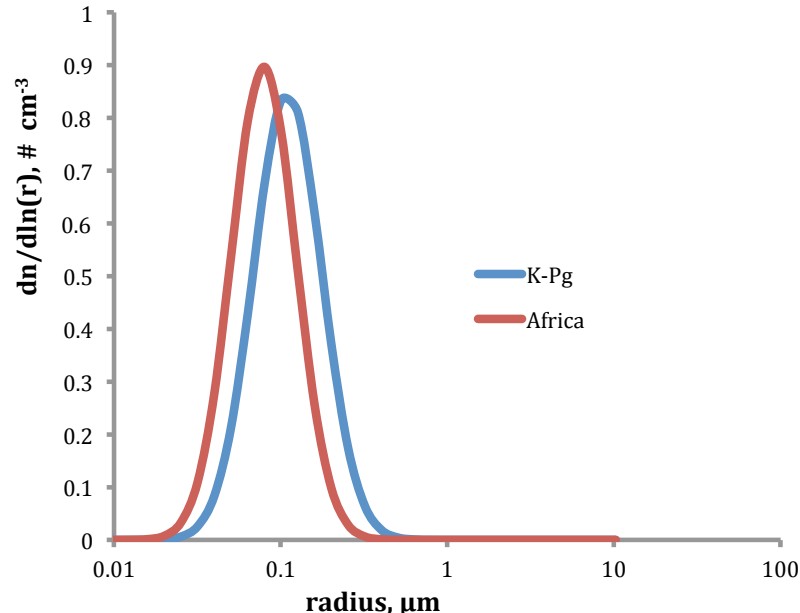

Fig. 2. The size distributions for smoke from modern fires in Africa, and from the K-Pg boundary layer (Wolbach et al., 1985; Matichuk et al., 2008)




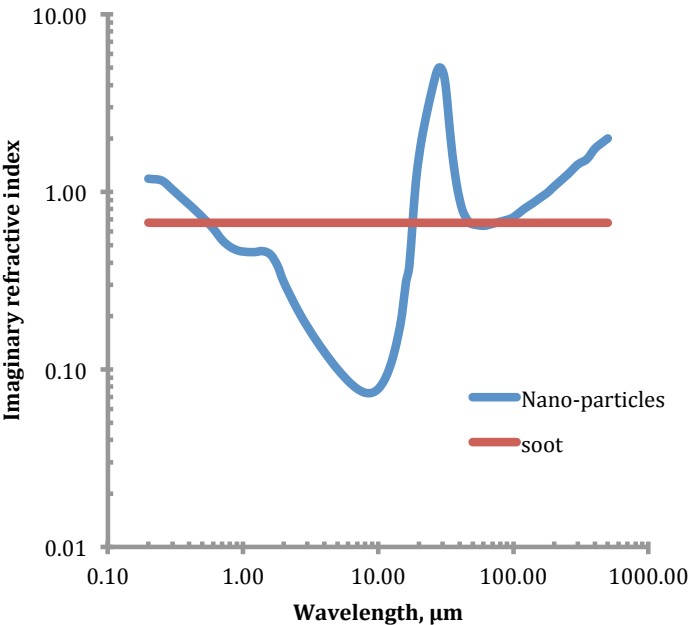

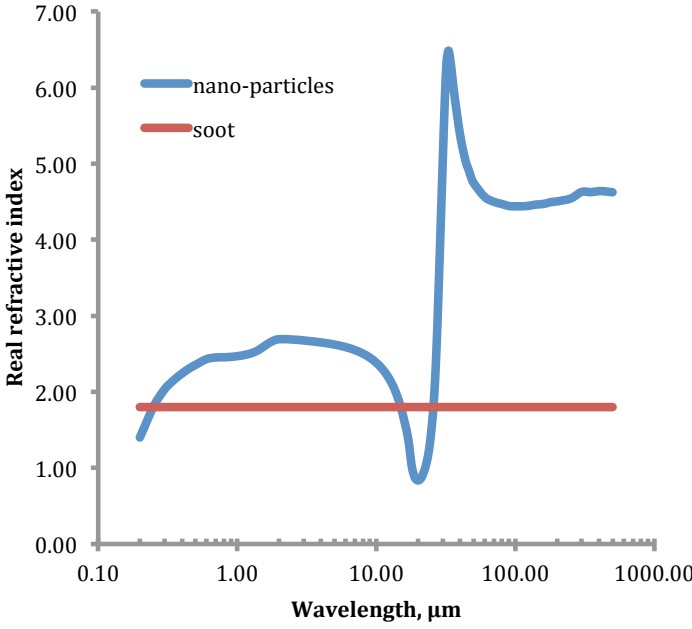

Fig. 3 The real and imaginary parts of the refractive index suggested for nano-particles, and for soot.