# Peer review of "Designing global climate and atmospheric chemistry simulations for km and"

_Atmospheric Chemistry and Physics, 2016_

## Referee Comment (RC1) · C. Covey (Referee) · 14 Jun 2016

As a (minor) coauthor of the Toon et al. 1997 paper, I'm pleased to see Toon's group return to this subject. Unfortunately I have neither participated nor closely observed progress in this field since 1997, so I'm unable to comment in detail. I do have the general impression that (1) some scenarios are quite robust despite the 65 million years that have passed since the K-Pg impact, e.g. lots of soot, and (2) the authors have done a thorough job laying out the initial conditions for a numerical weather / climate simulation. The initial conditions will serve not only the authors' future work but po-

tentially a large number of groups around the world that can perform such simulations. Hence this manuscript deserves publication in ACP.

---

## Referee Comment (RC2) · Anonymous Referee #1 · 15 Jun 2016

**1   General comments**

The paper contains a comprehensive review of material relevant to study the impact of asteroid strikes on the atmosphere of the Earth. It considers particulates of all scales released or formed during or after the collision including huge amounts of soot from ignited biomass which severely perturbs the radiation budget of the Earth, but also gases like halogens and sulfur perturbing atmospheric chemistry. This includes gases and particles released during a marine impact. The tables provide data to be used for simulations with comprehensive chemistry climate models. First simulations have

been performed by one of the authors using a state of the art model. The paper should be published in ACP after some minor corrections.

**2  Specific comments**

Sometimes the word "atmosphere" can be misleading. I suppose in line 414 troposphere is meant.

The sections 2.4.1.2 and 2.4.1.3 are difficult to understand because references are messed up or not listed. At the beginning of section 2.4.1.3 (line 836) this even leads to a sentence containing nonsense.

In general, some references are cited too often, sometimes words like "they" or "the latter" would be better.

Section 4 might be slightly shortened since it contains too many details discussed earlier. Better give a reference to WACCM at the end.

**3  Technical corrections**

Several citations have to be corrected because "et al" is missing or the paper is not listed or the author misspelled (lines 90, 310, 790, 830, 1085, 1096, 1104, 1213, 1328, 1332, 1335).

Typos in lines 792, 837(?), 879, 948, 1067, 1091, 1332, 1456, Table 5.

Insert "their" in line 418 before "Figure".

Something is missing in line 265.

I cannot find Hervig et al (2006), line 561, 588, 1698. Should it be Hervig et al (2009)?

Gulik et al (2008) is not cited.

Please complete citation in Table 1. Also better say there "< 1 $\mu$m". Improve caption or first 2 rows of Table 2 concerning units and scale factor. Ambient values or burden?

Table 3, row 2: total amount and column density?

Tables 2 and 4: The footnotes concerning $CO_2$ should be the same.

---

## Author Response (AR1)

**C. Covey (Referee)**

As a (minor) coauthor of the Toon et al. 1997 paper, I'm pleased to see Toon's group return to this subject. Unfortunately I have neither participated nor closely observed progress in this field since 1997, so I'm unable to comment in detail. I do have the general impression that (1) some scenarios are quite robust despite the 65 million years that have passed since the K-Pg impact, e.g. lots of soot, and (2) the authors have done a thorough job laying out the initial conditions for a numerical weather / climate simulation. The initial conditions will serve not only the authors' future work but potentially a large number of groups around the world that can perform such simulations. Hence this manuscript deserves publication in ACP.

**2. Authors' response**

We appreciate the time that the reviewers spent reading our manuscript and providing valuable suggestions for improving the paper.

Reviewer RC1, Curt Covey. Thank you for the comments. It does not appear that changes to the manuscript are needed.

**1. Comments from referees**

RC2: Interactive comment on "Designing global climate and atmospheric chemistry simulations for 1 km and 10, diameter asteroid impacts using the properties of ejecta from the K-Pg impact" by Owen B. Toon et al.

**Anonymous Referee #1**

**1** General comments**

The paper contains a comprehensive review of material relevant to study the impact of asteroid strikes on the atmosphere of the Earth. It considers particulates of all scales released or formed during or after the collision including huge amounts of soot from ignited biomass which severely perturbs the radiation budget of the Earth, but also gases like halogens and sulfur perturbing atmospheric chemistry. This includes gases and particles released during a marine impact. The tables provide data to be used for simulations with comprehensive chemistry climate models. First simulutions have been performed by one of the authors using a state of the art model. The paper should be published in ACP after some minor corrections.

**2 Specific comments**

Sometimes the word "atmosphere" can be misleading. I suppose in line 414 troposphere is meant.

The sections 2.4.1.2 and 2.4.1.3 are difficult to understand because references are messed up or not listed. At the beginning of section 2.4.1.3 (line 836) this even leads to a sentence containing nonsense.

In general, some references are cited too often, sometimes words like "they" or "the latter" would be better.

Section 4 might be slightly shortened since it contains too many details discussed earlier. Better give a reference to WACCM at the end.

**3** Technical corrections**

Several citations have to be corrected because "et al" is missing or the paper is not listed or the author misspelled (lines 90, 310, 790, 830, 1085, 1096, 1104, 1213, 1328, 1332, 1335).

Typos in lines 792, 837(?), 879, 948, 1067, 1091, 1332, 1456, Table 5. Insert "their" in line 418 before "Figure". Something is missing in line 265.

I cannot find Hervig et al (2006), line 561, 588, 1698. Should it be Hervig et al (2009)? C2

Gulik et al (2008) is not cited.

Please complete citation in Table 1. Also better say there "<  $1 \mu$ m". Improve caption or first 2 rows of Table 2 concerning units and scale factor. Ambient values or burden?

Table 3, row 2: total amount and column density?

Table 2 and 4: The footnotes concerning CO2 should be the same.

**2. Authors' response**

We appreciate the time that the reviewers spent reading our manuscript and providing valuable suggestions for improving the paper.

Reviewer RC2, General comments. Thank you for the synopsis. We don't see any corrections that are needed from the general comments.

Reviewer RC2, Specific comments:

**The reviewer's comment is listed and then our response is given in underlined italics.**

Sometimes the word "atmosphere" can be misleading. I suppose in line 414 troposphere is meant.

Yes, thank you, troposphere is what we meant.

The sections 2.4.1.2 and 2.4.1.3 are difficult to understand because references are messed up or not listed.

We altered a number of sentences in these sections to add references or clarify the discussion.

At the beginning of section 2.4.1.3 (line 836) this even leads to a sentence containing nonsense.

**We have corrected the line 836 as follows:**

The new application of the Pope (2002) approach leads to estimated submicron dust emissions that are about 500 times larger than the one derived by Pope (2002). The major difference is that we have assumed the ratio of quartz to clastics is about 1000, rather than 1 as assumed by Pope (2002).

In general, some references are cited too often, sometimes words like "they" or "the latter" would be better.

We were not able to identify where to make these changes. We would be happy to make additional changes if the reviewer could specify the passages that would benefit from revision.

Section 4 might be slightly shortened since it contains too many details discussed earlier.

We have not changed this section since some readers may read it without looking at the rest of the paper. It was also not clear to us what the reviewer would like to see changed.

Better give a reference to WACCM at the end.

We added:

We recently completed such simulations using the Whole Atmosphere Community Climate Model (WACCM) at the National Center for Atmospheric Research in a configuration similar to that used by Bardeen et al. (2008) and Mills et al. (2014).

**3** Technical corrections**

Several citations have to be corrected because "et al" is missing or the paper is not listed or the author misspelled (lines 90, 310, 790, 830, 1085, 1096, 1104, 1213, 1328, 1332, 1335).

**Thank you for noting these errors. We corrected them.**

Typos in lines 792, 837(?), 879, 948, 1067, 1091, 1332, 1456, Table 5. Insert "their" in line 418 before "Figure". Something is missing in line 265.

Thank you for noting these errors. We corrected them.

I cannot find Hervig et al (2006), line 561, 588, 1698. Should it be Hervig et al (2009)? C2

Yes, these should be (2009)

Gulik et al (2008) is not cited.

Thank you. This reference was not needed and has been removed.

Please complete citation in Table 1.

We assume this comment refers to the missing date in Orofino, which has been added.

Also better say there "< 1  $\mu$ m".

 $1\mu m$  is correct, this is a value needed for a rough calculation. We added the following at line 187

Particles smaller than 1µm would lead to a larger optical depth than given in Tables 1 and 3.

Improve caption or first 2 rows of Table 2 concerning units and scale factor. Ambient values or burden?

**We clarified the captions. Ambient burdens.**

Table 3, row 2: total amount and column density?

We clarified the captions. Ambient burdens.

Table 2 and 4: The footnotes concerning CO2 should be the same.

Corrected

In addition to these responses to the Reviewers' comments, we have also added material to the paper to update some numbers based on the work of Wolbach et al. (1990b). These reduce the magnitude of the soot emissions from Wolbach et al. (1988) by about 10% based on adding new data sets. We also slightly revised Table 1, to clarify the amount of soot that should be injected near the tropopause. This information was previously given in the text, but might have been missed by the reader. We clarified what to do if the injection into a model resulted in mixing ratios greater than 1. Finally, we added a few comments about a recent paper by Kaiho et al. (2016) who derived much different soot values than Wolbach et al. (1988, 1990b).

1

**Designing global climate and atmospheric chemistry simulations for 1 km and 10 km diameter asteroid impacts using the properties of ejecta from the K-Pg impact**

- 4 Owen B. Toon1, Charles Bardeen2, Rolando Garcia2
- 51 Department of Atmospheric and Oceanic Science, Laboratory for Atmospheric and
- 6 Space Physics, University of Colorado, Boulder
- 7 2 National Center for Atmospheric Research, Boulder, Colorado
- 8 *Correspondence to:* O.B. Toon (toon@lasp.colorado.edu)
- 9

10 Abstract. About 66 million years ago an asteroid about 10 km in diameter struck the Yucatan Peninsula creating the Chicxulub crater. The crater has been dated and found to 11 12 be coincident with the Cretaceous-Paleogene (K-Pg) mass extinction event, one of 6 great 13 mass extinctions in the last 600 million years. This event precipitated one of the largest 14 episodes of rapid climate change in Earth history, yet no modern three-dimensional 15 climate calculations have simulated the event. Similarly, while there is an on-going effort 16 to detect asteroids that might hit Earth and to develop methods to stop them, there have 17 been no modern calculations of the sizes of asteroids whose impacts on land would cause 18 devastating effects on Earth. Here we provide the information needed to initialize such 19 calculations for the K-Pg impactor and for a 1 km diameter impactor.

20 There is considerable controversy about the details of the events that followed the 21 Chicxulub impact. We proceed through the data record in the order of confidence that a 22 climatically important material was present in the atmosphere. The climatic importance 23 is roughly proportional to the optical depth of the material. Spherules with diameters of 24 several hundred-microns are found globally in an abundance that would have produced 25 an atmospheric layer with an optical depth around 20, yet their large sizes would only 26 allow them to stay airborne for a few days. They were likely important for triggering global wildfires. Soot, probably from global or near-global wildfires, is found globally in 27 28 an abundance that would have produced an optical depth near 100, which would 29 effectively prevent sunlight from reaching the surface. Nanometer sized iron particles are 30 also present globally. Theory suggests these particles might be remnants of the vaporized 31 asteroid and target that initially remained as vapor rather than condensing on the 32 hundred-micron spherules when they entered the atmosphere. If present in the greatest 33 abundance allowed by theory, their optical depth would have exceeded 1000. Clastics 34 may be present globally, but only the quartz fraction can be quantified since shock 35 features can identify it. However, it is very difficult to determine the total abundance of 36 clastics. We reconcile previous widely disparate estimates and suggest the clastics may 37 have had an optical depth near 100. Sulfur is predicted to originate about equally from 38 the impactor and from the Yucatan surface materials. By mass, sulfur is less than 10

39 percent of the observed mass of the spheres and estimated mass of nano-particles. Since

40 the sulfur probably reacted on the surfaces of the soot, nano-particles, clastics and 41 spheres, it is likely a minor component of the climate forcing; however, detailed studies

spheres, it is likely a minor component of the climate forcing; however, detailed studiesof the conversion of sulfur gases to particles are needed to determine if sulfuric acid

Microsoft Office User 8/11/2016 10:52 PM Deleted: S Microsoft Office User 8/11/2016 10:52 PM Deleted: diameter spherules

[revised manuscript text omitted]

Brian Toon 8/2/2016 12:39 PM Deleted: much Brian Toon 8/2/2016 12:39 PM Deleted: even

| DHah 10011 0/3/2010 10.23 AW                                                 |
|------------------------------------------------------------------------------|
| Deleted: ,                                                                   |
| Brian Toon 8/2/2016 9:50 AM                                                  |
| Deleted: (                                                                   |
| Brian Toon 8/2/2016 9:50 AM                                                  |
| Deleted: ),                                                                  |
| Brian Toon 8/2/2016 9:50 AM                                                  |
| Deleted: (                                                                   |
| Brian Toon 8/2/2016 9:50 AM                                                  |
| Deleted: ),                                                                  |
| Brian Toon 8/2/2016 9:50 AM                                                  |
| Deleted: . (                                                                 |
| Brian Toon 8/2/2016 9:50 AM                                                  |
| Deleted: )).                                                                 |
| Brian Toon 8/27/2016 1:12 PM                                                 |
| Deleted: and the other more recent papers                             |
| Brian Toon 8/27/2016 1:13 PM                                                 |
| Deleted: ,                                                                   |
| Brian Toon 8/3/2016 10:26 AM                                                 |
| Deleted: effect                                                              |
| Microsoft Office User 8/11/2016 11:27 PM                                     |
| Deleted: ,                                                                   |
| Microsoft Office User 8/11/2016 11:27 PM                                     |
| Deleted: and so must have remained in the atmosphere for a few years. |
| Brian Toon 8/2/2016 3:04 PM                                                  |
| Deleted: soot                                                                |
| Brian Toon 8/2/2016 4:11 PM                                                  |
| Deleted: 16                                                       |
| Brian Toon 8/2/2016 1:49 PM                                                  |
| Deleted:                                                                     |
| Brian Toon 8/2/2016 1:49 PM                                                  |
| Deleted: x10 -2                                                   |

|     |                                                                                                                                                                                   |                   | Deleted:                    |                |
|-----|-----------------------------------------------------------------------------------------------------------------------------------------------------------------------------------|-------------------|-----------------------------|----------------|
| 389 | undated these mass determinations to $5.6 \pm 1.5 \times 10^4$ Tg or $11\pm 3$ mg C cm -2 based on data                                                                |                   | Brian Toon 8/2/2016 4:11 F  | PM             |
| 390 | from 11 sites This mass of elemental carbon would require that the bulk of the above                                                                                              | $\leq$            | Formatted                   | [1]            |
| 391 | ground biomass burned and was partially converted to elemental carbon with an efficiency                                                                                          |                   | Brian Toon 8/2/2016 1:45 F  | PM             |
| 392 | of about 3% assuming the biomass is $1.5 \text{ g C} \text{ cm}^2$ of above ground dry organic mass per                                                                           | $\langle \rangle$ | Formatted                   | [2]            |
| 393 | $cm^2$ over the land area of Earth. This biomass density is typical of current tropical forests                                                                                   | $\langle \rangle$ | Brian Toon 8/2/2016 3:05 F  | PM             |
| 394 | This inferred 3% emission factor is about 60 times greater than that suggested by                                                                                                 | $\langle \rangle$ | Deleted: soot               |                |
| 305 | Andress and Marlet (2001) for current wildfires, but surges with laboratory and other                                                                                             |                   | Brian Toon 8/2/2016 3:11 F  | PM             |
| 396 | observations from huming wood under conditions consistent with mass fires (Crutzen et                                                                                             |                   | Deleted: soot               |                |
| 307 | al 1084: Turco et al 1000) Mass fires are more intense than forest fires and consume all                                                                                          |                   | Brian Toon 8/2/2016 3:12 F  | PM             |
| 200 | the fuel available possibly including that in the poor surface soil. Ivery and Salewitch                                                                                          |                   | Deleted: ,                  |                |
| 200 | the fuel available, possibly including that in the field surface soli. Ivally and Salawitch $(1002)$ argued independently from according earlier interval and solid $25\%$ of the |                   | Brian Toon 8/2/2016 3:12 F  | M              |
| 399 | (1993) argued independently from oceanic carbon isotope ratios that at least 23% of the                                                                                           |                   | Deleted: which              |                |
| 400 | above ground biomass must have burned at the K-rg boundary.                                                                                                                       |                   | Microsoft Office User 8/11/ | 2016 11:30 PM  |
| 401 | Well-shift $(1000h)$ distinguish second formula of elemental solution. A siniform radius is                                                                                       |                   | Deleted: The high soot elen | nental emi [3] |
| 40Z | woloach et al. (1990b) distinguish several forms of elemental carbon. Aciniform carbon is                                                                                         |                   | Brian Toon 8/2/2016 3:23 F  | PM             |
| 403 | composed of grape-like clusters of 0.01 to 0.1 $\mu$ m spherules. On average, this type of soot                                                                                   |                   | Formatted                   | [4]            |
| 404 | is 20.0% of the elemental carbon, yielding a global mass abundance of 1.5x10, 1g of                                                                                               |                   | Brian Toon 8/2/2016 3:29 F  | PM             |
| 405 | aciniform carbon. Charcoal is estimated at 3.3 to 4.1x10 Ig, and unreactive kerogen at 0                                                                                          |                   | Formatted                   | [5]            |
| 406 | to 0.8x10 -1 Ig. Wolbach et al. (2003) discuss a data set from the mid-Pacific that suggests                                                                           |                   | Brian Toon 8/2/2016 3:29 F  | M              |
| 407 | aciniform soot is 9x10° 1g, and charcoal is also 9x10° 1g. Wolbach et al. directly measure                                                                                        |                   | Formatted                   | [6]            |
| 408 | the carbon content of their samples. The aciniform soot to charcoal ratio is determined by                                                                                        | $\backslash$      | Brian Toon 8/2/2016 4:34 F  | PM             |
| 409 | using an electron microscope to distinguish small and large particles.                                                                                                            |                   | Formatted                   | [7]            |
| 410 |                                                                                                                                                                                   |                   | Brian Toon 8/2/2016 4:34 F  | M              |
| 411 | There are several uncertainties in determining the amount of soot to use in a model. An                                                                                           |                   | Formatted                   | [8]            |
| 412 | upper limit of the amount injected into the stratosphere is 7.1 x104 Tg based on the upper                                                                      |                   | Brian Toon 8/2/2016 4:50 F  | PM             |
| 413 | error bar of the Wolbach et al. (1990b) elemental carbon values. An important assumption                                                                                          |                   | Formatted                   | [9]            |
| 414 | in this upper limit is that the larger particles found by Wolbach et al. (1990b), are either                                                                                      |                   | Brian Toon 8/2/2016 4:55 F  | PM             |
| 415 | aggregates of smaller ones, or of the same general size as the aggregates of the smaller ones                                                                                     |                   | Formatted                   | [10]           |
| 416 | that occur after coagulation. A lower limit of $1.1 \times 10^4$ Tg is obtained using the lower error                                                                             | / /               | Microsoft Office User 8/11/ | 2016 11:02 PM  |
| 417 | bar of the elemental carbon from Wolbach et al. (1990b), and assuming 26.6% is aciniform                                                                                          |                   | Deleted: 0                  |                |
| 418 | soot. Alternatively, one could argue that this lower limit of aciniform soot should be                                                                                            |                   | Brian Toon 8/2/2016 5:03 F  | PM             |
| 419 | injected into the stratosphere, along with 3.3x10 4 Tg of charcoal using different size                                                                                |                   | Formatted                   | [11]           |
| 420 | distributions. The most likely value of the aciniform soot in the stratosphere is $1.5 \times 10^4$ Tg,                                                                           |                   | Brian Toon 8/3/2016 10:44   | AM             |
| 421 | and of elemental carbon 5.6x10 4 Tg. We use these most likely values in Table 1.                                                                                       |                   | Formatted                   | [12]           |
| 422 |                                                                                                                                                                                   |                   | Brian Toon 8/3/2016 10:44   | AM             |
| 423 | Kaiho et al. (2016) argue that the soot came from burning hydrocarbons in the crater and                                                                                          |                   | Formatted                   | [13]           |
| 424 | that the total mass emitted was either $5 \times 10^2$ , $15 \times 10^2$ or $26 \times 10^2$ Tg. If we reduce these                                                              |                   | Brian Toon 8/3/2016 10:40   | AM             |
| 425 | values by the author's factor of 2.6 to represent the stratospheric emissions, they are 0.4%,                                                                                     |                   | Formatted                   | [14]           |
| 426 | 1.0% and 1.7% of the globally distributed elemental carbon reported by Wolbach et al.                                                                                             |                   | Brian Toon 8/3/2016 10:40   | AM             |
| 427 | (1990b).                                                                                                                                                                   |                   | Formatted                   | [15]           |
| 428 |                                                                                                                                                                                   |                   | Brian Toon 8/3/2016 10:40   | AM             |
| 429 | Kaiho et al. (2016) measured several polycyclic aromatic hydrocarbons (PAHs) that are                                                                                             |                   | Formatted                   | [16]           |
| 430 | minor components of soot from one distal site in Caravaca, Spain, and another site at                                                                                             |                   | Microsoft Office User 8/11/ | 2016 11:02 PM  |
| 431 | Beloc, Haiti that is about 700 km from the crater. Since the PAHs measured are minor                                                                                              |                   | Deleted: they               |                |
| 432 | constituents of soot Kaiho et al. (2016) need to use a large correction factor to determine                                                                                       |                   | Microsoft Office User 8/11/ | 2016 11:03 PM  |
| 433 | the amount of soot. They first multiply by factors of 2, 5.9, or 10 to account for possible                                                                                       |                   | Deleted: re                 |                |
| 434 | loss of PAH concentrations over time. They presented no data to justify these factors. They                                                                                       |                   | Microsoft Office User 8/11/ | 2016 11:03 PM  |
|     |                                                                                                                                                                                   |                   | Deleted: were no data       |                |

Brian Toon 8/2/2016 1:43 PM

[revised manuscript text omitted]

**Brian Toon 8/3/2016 8:02 PM Deleted: is Brian Toon 8/3/2016 7:59 PM Formatted: Left, Indent: Left: 0.5", First line: 0", Space Before: 0 pt Microsoft Office User 8/12/2016 2:55 PM Deleted: (2) Brian Toon 8/3/2016 7:56 PM Deleted: Brian Toon 8/3/2016 7:59 PM Formatted: Font: Times Brian Toon 8/3/2016 7:56 PM Formatted: Font: Times, Not Italic Brian Toon 8/28/2016 9:08 AM Deleted: T Brian Toon 8/3/2016 8:58 PM Deleted: I =**

Brian Toon 8/3/2016 7:59 PM

Brian Toon 8/3/2016 8:00 PM

Brian Toon 8/3/2016 8:07 PM **Deleted:** *I*T

Microsoft Offic Deleted: (3)

Brian Toon 8/3/2016 7:55 PM

Brian Toon 8/3/2016 7:55 PM

Brian Toon 8/3/2016 8:07 PM Deleted: is

[revised manuscript text omitted]

Microsoft Office User 8/11/2016 11:06 PI Deleted: Microsoft Office User 8/11/2016 11:06 PM Deleted: push it upward

Brian Toon 8/2/2016 10:55 AM Deleted: calculate Brian Toon 8/2/2016 10:56 AM Deleted: in the rising fireball

- 696 mixture of impactor and asteroid, so the 44% mass fraction is approximately equal to the
- 697 mass of the impactor. This 44% vapor fraction depends on the pressures reached in the
- 698 impact, the equation of state of the materials, as well as the detailed evolution of the
- debris in the fireball. The fate of this vapor phase material is not well understood, and has

700 been little studied. It may simply have condensed on the spherules, or it may have

701 remained as vapor.

702 Presently, 100  $\mu$ m and larger sized micro-meteoroids ablate to vapor in the upper 703 atmosphere. Hunten et al. (1980), following earlier suggestions, modeled the 704 condensation of these rock vapors as they form nm-sized particles in the mesosphere and 705 stratosphere. Bardeen et al. (2008) produced modern models of their distribution based 706 on injection calculations from Kalashnikova et al. (2000). Hervig et al. (2009) and Neely 707 et al. (2011) showed that these tiny particles are observed as they deposit about 40 tons of 708 very fine-grained material on Earth's surface per day. It is possible that a similar process 709 occurred after the Chicxulub impact. However, in the Chicxulub case the vaporization 710 occurred during the initial asteroid impact at Chicxulub rather than on reentry of the 711 material after the fireball rose thousands of km into space and dispersed over the globe.

inaternal arter the medan rose mousands of Kin into space and dispersed over the glob

The presence of 15-25 nm diameter, iron-rich material has been recognized in the fireball layer at a variety of sites by Wdowiak et al. (2001), Verma et al. (2002), Bhandari et al. (2002), Ferrow et al. (2011) and Vajda et al. (2015) among others. The nano-phase iron correlates with iridium, is found worldwide, and therefore is likely a product of the impact process. Unfortunately, these authors have not quantified the amount of this material that is present. Berndt et al. (2011) were able to perform very high-resolution

718 chemical analyses, and also report a component of the platinum group elements that 719 arrived later than the bulk of the ejecta, and was probably the result of submicron sized

- particles. However, they were not able to size the particles, nor quantify their abundance.
- In Table 1 we take the upper limit of the injected mass of nano-particles to be  $2 \ge 10^{18}$  g. 721 722 The lower limit is zero. This choice for the upper limit is consistent with the vapor mass 723 Jeft at the end of the simulations by Johnston and Melosh (2012b). We assume an initial 724 diameter of 20 nm, following Wdowiak et al. (2001). We assume the particles are 725 initially injected over the same altitude range as the Type 2 spherules, because we 726 speculate that the small particles would not separate from the bulk of the ejecta in the 727 fireball until the ejecta entered the atmosphere and reached terminal velocity. The mass 728 injected would lead to an optical depth of particles larger than 1000 even if they coagulated into the 1  $\mu$ m size range. Goldin and Melosh (2009) point out that such an 729 730 optically thick layer of small particles left behind by the falling large spheres might also 731 be important for determining whether the infrared radiation from the atmosphere heated 732 by the Type 2 spherules is sufficient to start large-scale fires.

733 The optical properties of the nano-particles are not known. We suggest using the optical

properties of the small, vaporized particles currently entering the atmosphere from Hervig

et al. (2009). These optical constants are plotted in Figure 3. We also assume that the

particles have the density of CM2 asteroids, since Cr isotope ratios suggest that is the

composition of the K-Pg impactor (Trinquier et al., 2006). This density is 2.7 g cm3. A
 significant fraction of the vaporized material may be from the impact site, so using an

739 asteroidal composition to determine the density is an approximation.

Brian Toon 8/2/2016 10:02 AM Deleted: 6 Brian Toon 8/2/2016 9:58 AM Deleted: Brian Toon 8/2/2016 10:58 AM Deleted: likely

Brian Toon 8/2/2016 11:01 AM **Deleted:** estimate

Brian Toon 8/2/2016 10:03 AM Deleted: 6

**745**

[revised manuscript text omitted]

Brian Toon 8/2/2016 10:04 AM

structure, Nature Geosci., 1, 131-135, 2008.

| Brian Toon 8/3/2016 7:44 PM                                                   |
|-------------------------------------------------------------------------------|
| Formatted: Justified, Indent: Left: 0",
Hanging: 0.25", Space Before: 6 pt |
| Brian Toon 8/3/2016 7:42 PM                                                   |
| Formatted: Font:Times, 12 pt                                                  |
| Brian Toon 8/3/2016 7:42 PM                                                   |
| Formatted: Font:Times, 12 pt                                                  |
| Brian Toon 8/3/2016 7:42 PM                                                   |
| Formatted: Font:Times, 12 pt                                                  |
| Brian Toon 8/3/2016 7:42 PM                                                   |
| Formatted: Font:Times, 12 pt                                                  |

[revised manuscript text omitted]

(0.44)                                       | $\frac{1.5-5.6 \text{ x}}{10^{16} (0.29)}$ $\frac{\text{to } 1.1 \text{ x} 10^{-2}}{2 \text{ x}^{\text{***}}}$ | ~2x10 18**
(0.4)          | <6x10 16
(0.01)                                    | $\frac{9 \times 10^{16}}{(5.4 \times 10^{-2} \text{ g})}$ |
| Global
optical
depth as
$1 \mu m$
particles *                          | ~20 (for
250 µm
particles)                                      | ~100                                                                                                           | ~2000                                   | ~90                                                              | ~450                                                      |
| Vertical
distribution                                                           | 70 km_
Gaussian
distribution
with half
width of
6.6 km | Eq. 2,                                                                                                         | Same as
Type 2
spherules          | Uniformly
mixed
vertically
above
tropopause          | Same as
Type 2
spherules                            |
| Optical properties                                                                 | Not
relevant                                                       | n=1.8
k=0.67                                                                                                | Hervig et
al., (200 <mark>9</mark> ) | Orofino et
al. (1998)
Jimestone                            | Sulfuric acid                                             |
| Initial
Particle
size                                                        | 250 μm
diameter                                                    | Lognormal,
$r_{m}=0.11\mu$ m,
$\sigma=1.6$ ;
monomers
30-60 nm                                     | 20 nm
diameter                       | Lognormal,
$\underline{r}_m = 0.5 \mu m$ ,
$\sigma = 1.65$ | gas                                                       |
| Material
density, g
cm -3                                         | 2.7                                                                   | 1.8                                                                                                            | 2.7                                     | 2.7                                                              | 1.8                                                       |

\*Qualitative estimate for comparison purposes only 1899

\*This value is an upper limit. The lower limit is zero 1900

These values are for aciniform soot, or elemental carbon in the stratosphere, see text. 1901

\*\*\*\*The material may have quickly moved to below 50 km to maintain hydrostatic 1902 balance. See text.

1903 1904

| Brian Toon 8/3/2016 7:51 PM    |
|--------------------------------|
| Deleted: from vaporized rock   |
| Brian Toon 8/27/2016 9:02 PM   |
| Deleted: distributed globally  |
| Brian Toon 8/3/2016 7:44 PM    |
| Deleted: 7                     |
| Brian Toon 8/3/2016 7:45 PM    |
| Deleted: 3                     |
| Brian Toon 8/2/2016 10:20 AM   |
| Deleted: (                     |
| Brian Toon 8/3/2016 7:50 PM    |
| Formatted: Superscript         |
| Brian Toon 8/2/2016 10:20 AM   |
| Deleted: )                     |
| Brian Toon 8/27/2016 9:03 PM   |
| Deleted: Estimated             |
| Brian Toon 8/27/2016 9:03 PM   |
| Deleted: g                     |
| Brian Toon 8/3/2016 7:45 PM    |
| Deleted: Eq. 2                 |
| Brian Toon 8/27/2016 9:04 PM   |
| Deleted: as center of          |
| Brian Toon 8/27/2016 9:01 PM   |
| Formatted: Superscript         |
| Brian Toon 8/2/2016 10:03 AM   |
| Deleted: 6                     |
| Brian Toon 8/27/2016 9:04 PM   |
| Deleted: for                   |
| Brian Toon 8/28/2016 8:49 AM   |
| Deleted: modal particle radius |
| Brian Toon 8/28/2016 8:50 AM   |
| Deleted: modal particle radius |
| Brian Toon 8/28/2016 8:49 AM   |
| Formatted: Subscript           |
| Brian Toon 8/28/2016 8:49 AM   |
| Formatted: Font:Symbol         |
| Brian Toon 8/28/2016 8:49 AM   |
| Deleted: sigma                 |
| Brian Toon 8/28/2016 8:50 AM   |
| Deleted: sigma                 |
| Brian Toon 8/2/2016 11:03 AM   |
|                                |
| Brian Toon 8/3/2016 7:52 PM    |
|                                |
| Brian Toon 8/27/2016 9:03 PM   |
|                                |

1921

the Chivenluh in 1922 . . ()2.0

| s                 | C (as                                                                                                                                                                                  | но                                                                                                                                                       | Cl                                                                                                                                                                                                                                                                                                                                                               | Br                                                                                                                                                                                                                                                                  | T                                                                                                                                                                                                                                                                                                                                                                                             | N                                                                                                                                                                                                                                                                    | Vertical                                                                                                                                                                                                                                                                                                                                                                                                                                                                                                                                                                                                                                                                                                                                                                                                                                                                                                                                                                                                                                                                                                                                                                                                                                                                                                                                                                                                                                                                             |
|-------------------|----------------------------------------------------------------------------------------------------------------------------------------------------------------------------------------|----------------------------------------------------------------------------------------------------------------------------------------------------------|------------------------------------------------------------------------------------------------------------------------------------------------------------------------------------------------------------------------------------------------------------------------------------------------------------------------------------------------------------------|---------------------------------------------------------------------------------------------------------------------------------------------------------------------------------------------------------------------------------------------------------------------|-----------------------------------------------------------------------------------------------------------------------------------------------------------------------------------------------------------------------------------------------------------------------------------------------------------------------------------------------------------------------------------------------|----------------------------------------------------------------------------------------------------------------------------------------------------------------------------------------------------------------------------------------------------------------------|--------------------------------------------------------------------------------------------------------------------------------------------------------------------------------------------------------------------------------------------------------------------------------------------------------------------------------------------------------------------------------------------------------------------------------------------------------------------------------------------------------------------------------------------------------------------------------------------------------------------------------------------------------------------------------------------------------------------------------------------------------------------------------------------------------------------------------------------------------------------------------------------------------------------------------------------------------------------------------------------------------------------------------------------------------------------------------------------------------------------------------------------------------------------------------------------------------------------------------------------------------------------------------------------------------------------------------------------------------------------------------------------------------------------------------------------------------------------------------------|
| $(x10^{13})$      | $C(as) CO_2^{**}$                                                                                                                                                                      | $(\underline{x} 10^{15})$                                                                                                                                | $(\underline{x}^{10^{12}})$                                                                                                                                                                                                                                                                                                                                      | $(\underline{x}^{10^{10}})$                                                                                                                                                                                                                                         | $(x 10^7)$                                                                                                                                                                                                                                                                                                                                                                                    | $(\underline{x}10^{14})$                                                                                                                                                                                                                                             | distribution                                                                                                                                                                                                                                                                                                                                                                                                                                                                                                                                                                                                                                                                                                                                                                                                                                                                                                                                                                                                                                                                                                                                                                                                                                                                                                                                                                                                                                                                         |
| 1*                | 8.4                                                                                                                                                                                    | 1.3
strat                                                                                                                                             | 2.3
strat                                                                                                                                                                                                                                                                                                                                                     | 3.1
strat                                                                                                                                                                                                                                                        | <2.3
strat                                                                                                                                                                                                                                                                                                                                                                                 | 2
as N 2 O                                                                                                                                                                                                                                             |                                                                                                                                                                                                                                                                                                                                                                                                                                                                                                                                                                                                                                                                                                                                                                                                                                                                                                                                                                                                                                                                                                                                                                                                                                                                                                                                                                                                                                                                                      |
| 4x10 3 | 0.3                                                                                                                                                                                    | 200                                                                                                                                                      | 7x10 2                                                                                                                                                                                                                                                                                                                                                | 5x10 2                                                                                                                                                                                                                                                   | 7x10 4                                                                                                                                                                                                                                                                                                                                                                             |                                                                                                                                                                                                                                                                      | As Type 2
spherules                                                                                                                                                                                                                                                                                                                                                                                                                                                                                                                                                                                                                                                                                                                                                                                                                                                                                                                                                                                                                                                                                                                                                                                                                                                                                                                                                                                                                                                               |
| 40                | 6                                                                                                                                                                                      | 1500                                                                                                                                                     | 200                                                                                                                                                                                                                                                                                                                                                              | 1000                                                                                                                                                                                                                                                                | 9x10 5                                                                                                                                                                                                                                                                                                                                                                             | 10                                                                                                                                                                                                                                                                   | As soot